SciPost Physics

# Localization and fractality in disordered Russian Doll model

Vedant R. Motamarri[1,2], Alexander S. Gorsky[3,4], and Ivan M. Khaymovich[1,5,6,*]

**1** Max-Planck-Institut für Physik komplexer Systeme, Nöthnitzer Straße 38, 01187-Dresden, Germany
**2** Indian Institute of Technology Bombay, Mumbai 400076, India
**3** Institute for Information Transmission Problems RAS, 127051 Moscow, Russia
**4** Moscow Institute for Physics and Technology, Dolgoprudny 141700, Russia
**5** Institute for Physics of Microstructures, Russian Academy of Sciences, 603950 Nizhny Novgorod, GSP-105, Russia
**6** Nordita, Stockholm University and KTH Royal Institute of Technology Hannes Alfvéns väg 12, SE-106 91 Stockholm, Sweden
* ivan.khaymovich@gmail.com

September 15, 2023

## Abstract

Motivated by the interplay of Bethe-Ansatz integrability and localization in the Richardson model of superconductivity, we consider a time-reversal symmetry breaking deformation of this model, known as the Russian Doll Model (RDM), and implement diagonal on-site disorder. The localization and ergodicity-breaking properties of the single-particle spectrum are analyzed using a large-energy renormalization group (RG) over the momentum-space spectrum. Based on the above RG, we derive an effective Hamiltonian of the model, discover a fractal phase of non-ergodic delocalized states – with the fractal dimension different from the paradigmatic Rosenzweig-Porter model – and explain it in terms of the developed RG equations and the matrix-inversion trick.

# 1   Introduction

The Richardson model of superconductivity [1, 2] is a suitable toy model with a finite number of degrees of freedom which captures the key properties of the superconducting state in a relatively simple manner. This model, given by on-site potential $\varepsilon_n$ on $N$ sites and all-to-all constant coupling $j_{mn} = const/N$, is known to be Bethe-Ansatz (BA) integrable, where the BA equations coincide with the ones for the twisted SU(2) Gaudin model [3]. The commuting integrals of motion (Hamiltonians) emerging from BA in the Richardson model get identified as superpositions of the Gaudin Hamiltonians.

The relation between integrability and localization properties of the Richardson model with diagonal disorder has been considered in [4, 5] in the single-particle sector of the model, where it was shown that all (except one) eigenstates are localized for any coupling constant $j_{mn} \ll N^{-1}$ ($j_{mn} \gg N^{-1}$). The delocalization of the only level appears at the same coupling, $j_{mn} \simeq N^{-1}$, at which the superconducting gap in the many-body sector starts to become extensive. Though all (except one) eigenstates are localized for any coupling due to the BA integrability, the corresponding level statistics shows level repulsion for $j_{mn} > N^{-1}$, which is comparable with the one in the random matrix theory of Gaussian random ensembles [6]. This indicates the non-trivial relation of BA integrability to the localization properties already at the single-particle level.

As Anderson localization, based on interference effects, is highly sensitive to the magnetic field, it is of particular interest to go beyond the Richardson model by breaking time-reversal symmetry but at the same time retaining the BA integrability. Such an integrable deformation of the Richardson model is the so-called Russian Doll model (RDM) [7, 8] Similar to the Richardson model, the RDM has all-to-all constant coupling $j_{mn} = [g + ih\text{sign}(m-n)]/N$, but apart from the symmetric real term $\sim g$, it also has an antisymmetric imaginary contribution $\sim ih\text{sign}(m-n)$. In this case, the BA equations coinicide with the twisted inhomogeneous XXX $SU(2)$ spin chain. The inhomogeneous magnetic field in the latter model is associated with the on-site potential in RDM, while the twist is the counterpart of the coupling constant. The TRS breaking parameter $h$ in RDM is identified as the "Planck constant" in XXX spin chain which vanishes in the Gaudin limit [9]. This model can be also related to Chern-Simons theory where the excitations are represented by the vertex operators [10]. The RDM serves as an example of a cyclic RG, where the TRS breaking parameter provides the period of the cycle, see the

review [11].

In this paper, motivated by the interplay of the BA integrability, localization, and level repulsion in the Richardson model, we consider the Russian Doll model bringing TRS breaking into the game, along with diagonal disorder. As in previous localization studies of the Richardson model, we focus on the single-particle sector of the RDM, which has much in common with the many-body physics including the tower of high-energy ground state solutions. We also consider the generalization of RDM in terms of scaling of the coupling constant. In the original RDM, the coupling scales as $N^{-1}$, while we consider more general scaling $N^{-\gamma/2}$ in analogy with the Rosenzweig-Porter model [12]. The latter model also consists of all-to-all hopping terms, but the couplings are given by i.i.d. Gaussian random variables with the standard deviation $N^{-\gamma/2}$.

The Rosenzweig-Porter model is known to host an entire phase of non-ergodic (so-called fractal) eigenstates in the range $1 < \gamma < 2$, squeezed between the ergodic ($\gamma < 1$) and Anderson localized ($\gamma > 2$) phases [13]. The non-ergodic phase is characterized by the only energy scale $\Gamma$, large compared to the level spacing $\delta \sim 1/[N\rho(E)]$ and small compared to the bandwidth $\sim 1/\rho(E)$ of the spectrum, where $\rho(E)$ is the density of states. This energy scale is given by the standard Fermi's Golden rule

$$\Gamma = \frac{2\pi}{\hbar}\rho(E)\sum_m |j_{mn}|^2 \sim \delta N^D \ , \tag{1}$$

and it determines the fractal dimension $0 < D = 2 - \gamma < 1$ of the wave-function support set. Later, several other models with similar fractal [14–18] and multifractal [19–24] phases have been suggested in the literature. In all these models, it has been shown that the wave-function structure is determined mostly by the diagonal elements, while the hopping terms provide a certain Breit-Wigner level broadening $\Gamma$. [1].

For the Richardson model, the standard Fermi's Golden rule result fails to describe the localization properties correctly due to the presence of strong correlations between the coupling of different sites. In the case of localization – which survives for any coupling strength even beyond the convergence of the locator expansion (like in the Richardson model [4,5] and other long-range fully correlated models [26,27]) – one can use a so-called matrix-inversion trick [15] or develop a strong-disorder spatial RG [28,29].

For RDM, in this work we show that increasing the coupling does lead to the delocalization of most of the eigenstates, and therefore, both the above methods that work only in the localized phase, are not applicable. At the same time, the standard Fermi's Golden rule approximation (1) fails due to the strongly correlated coupling terms. Therefore, our goal here is to develop another analytical method to describe localization and ergodicity-breaking properties of RDM. We base our approach on the RG flow, similar in spirit to the one used for disorder-free RDM for $\gamma = 2$ [7], but generalize it to the momentum space. In order to double-check the RG approximations, we also generalize the above-mentioned matrix-inversion trick to the case of any unbound spectrum of the disorder-free coupling term $j_{mn}$, and show that the effective Hamiltonian obtained by this method is statistically equivalent to the one calculated from the RG.

By going back to the coordinate basis, we derive the effective Hamiltonian with significantly reduced correlations, which is readily tractable with the Fermi's Golden rule approximation (1). The effective Hamiltonian makes it possible to elaborate the localization properties of the single-particle states. Using a combination of the effective Hamiltonian and Fermi's Golden rule, we find that single-particle eigenstates in the disordered RDM demonstrate fractal properties, emerging at the same Anderson localization point $\gamma = 2$

---

[1]Moreover this works also for the non-Hermitian Rosenzweig-Porter model [25], where the phase diagram is affected only by the non-Hermiticity of the diagonal matrix entries, but not by hopping terms.

as the Rosenzweig-Porter model. However, the non-ergodic phase is prolonged to smaller values of $\gamma$, i.e. $\gamma = 0$, and the corresponding fractal dimension $D$, which we determine exactly analytically, also deviates from the one in the Rosenzweig-Porter model and equals to $D = 1 - \gamma/2$.

The remainder of the paper is organized as follows. In Sec. 2 we explicitly describe the disordered Russian Doll model. Next, in Sec. 3 we calculate the spectrum of the disorder-free RDM, describe it in terms of energy stratification [17], and calculate the localization properties of the energy-stratified states in the momentum basis. In Sec. 4 we derive an effective Hamiltonian representation for RDM using a high-energy RG in the momentum space. Section 5 represents the generalization of the matrix-inversion trick introduced in [15] in order to make it applicable to the description of the delocalized states and confirm its equivalence to the above RG by comparing the results for the effective Hamiltonian. In Sec. 6 we provide the analytical results leading from the structure of the effective Hamiltonian, supported with numerical simulations. The conclusion and outlook are given in Sec. 7.

## 2   Model

In this work, we focus on the single-particle sector of the Russian Doll model with on-site disorder $\varepsilon_n$ and generalized coupling amplitude $j_{mn} \sim N^{-\gamma/2}$. The single-particle Hamiltonian in the coordinate basis is the $N \times N$ random matrix

$$H_{mn} = \delta_{mn}\varepsilon_n - j_{mn}, \quad j_{mn} = \frac{g + ih\,\mathrm{sign}\,(d(m,n))}{N^{\gamma/2}} \ . \tag{2}$$

where $1 \leq m,n \leq N$. Here, the generalized coupling $j_{mn}$ scales as power $-\gamma/2$ of the system size $N$, as opposed to $N^{-1}$, and the on-site potentials $\varepsilon_n$ are given by Gaussian i.i.d. variables

$$\langle \varepsilon_n \rangle = 0, \quad \langle \varepsilon_n^2 \rangle = W^2 \ . \tag{3}$$

The above-mentioned symmetric coupling $g$ and the TRS breaking parameter $h$ are parameterized by the angular variable $\theta$ as follows

$$g = \cos\theta, \quad h = \sin\theta \ , \quad 0 \leq \theta < 2\pi \ . \tag{4}$$

For simplicity, we consider periodic boundary conditions and define the distance $d(m,n)$ between sites $m$ and $n$ with a sign: if the shortest route from $m$ to $n$ is clockwise (counterclockwise), the distance is positive (negative), see Fig. 1,

$$d(m,n) = (m-n) \mod N, \quad |m-n| \leq N/2 \ . \tag{5}$$

This allows us to determine an effective magnetic flux $\theta$, threading the loop $m - n - m$ and equal for each link between any pair of sites $m$ and $n$.

Note that according to Anderson resonance counting [13,15,30,31], a general principle of Anderson localization in long-range models, a measure one subset of the states in this model are localized for $\gamma > 2$, irrespective of *any* correlations or TRS breaking. Consequently, all the properties present in the Richardson model or Rosenzweig-Porter model at $\gamma > 2$ – such as the Lorentzian power-law profile of the eigenstates versus $\varepsilon_i$ (sometimes called frozen multifractality) and the power-law Chalker scaling of the wave-function overlap, absent in the short-range Anderson models – are also present in the disordered Russian Doll model. Therefore, unless mentioned otherwise, we focus on the range $0 < \gamma < 2$ in the further sections.

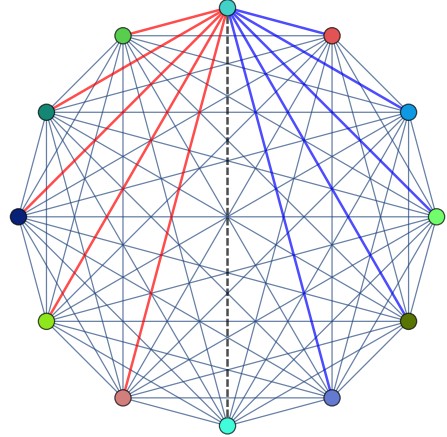

Figure 1: **Sketch of the Russian doll model, Eqs.** (2)-(4)**.** Different colors of vertices stand for the disorder potential $\varepsilon_n$, while the coloring of the edges from the topmost vertex demonstrate different phases of hopping terms with the same amplitude: red color stands for $e^{i\theta}$, blue – for $e^{-i\theta}$, and black dashed line corresponds to the real hopping 1.

## 3  Energy stratification of the spectrum and ergodicity-breaking in momentum space

In this section, we focus on the spectrum of the disorder-free RDM. Using the property of its stratification, we analyze the localization and the ergodicity-breaking properties of large-energy eigenstates of the corresponding disordered RDM. Indeed, the coupling/hopping matrix $j$ is translation invariant $j_{mn} = j_{m-n}$ and can be thus diagonalized in the basis of plane waves

$$|p\rangle = \sum_n \frac{e^{\frac{2\pi inp}{N}}}{\sqrt{N}} |n\rangle \ , \tag{6}$$

with the spectrum indexed by an integer $|p| \leq N/2$

$$E_0 = N^{1-\gamma/2} \cos\theta \tag{7a}$$

$$E_{2k\neq0} = \begin{cases} 0, & \text{even } N \\ -N^{-\gamma/2} \sin\theta \tan\left(\frac{\pi k}{N}\right), & \text{odd } N \end{cases} \tag{7b}$$

$$E_{2k+1} = \begin{cases} 2N^{-\gamma/2} \sin\theta \cot\left(\frac{\pi(2k+1)}{N}\right), & \text{even } N \\ N^{-\gamma/2} \sin\theta \cot\left(\frac{\pi(2k+1)}{2N}\right), & \text{odd } N \end{cases} \tag{7c}$$

From this spectrum one can immediately see that

- For the Richardson model ($\theta = 0$), the spectrum is $(N-1)$-fold degenerate, $E_{p\neq0} = 0$, with the only non-zero energy level $E_0 \sim N^{1-\gamma/2}$. It is this level which is responsible for the localization of the other $N-1$ eigenstates orthogonal to it in the disordered Richardson model for $\gamma < 2$ [4,5].

- Even in the general case of $\theta \neq 0$, the levels with non-zero and even $p = 2k$ remain small, $|E_{2k}| < N^{-\gamma/2}$, for odd $N$, and are zero for even $N$. Later, we focus on the case with even $N$ to neglect the small amplitude of these levels. However, the levels with odd $p = 2k + 1$, for any finite $\theta$, are as significant as $E_0$. For $|k| \ll N$ one can

write

$$E_{2k+1} \sim \sin\theta \frac{2N^{1-\gamma/2}}{\pi(2k+1)} \ . \tag{8}$$

Note that the transition between Richardson model and RDM (see the above cases) occurs for the TRS breaking parameter $\theta_c \sim W N^{-(1-\gamma/2)}$. For $\theta < \theta_c$, the largest of the energy levels $E_{2k+1}$, namely $E_1$, becomes smaller than the diagonal disorder amplitude $W$ and thus becomes hybridized with the rest of the zero modes by the disorder. As $N \to \infty$, this transition occurs at $\theta_c \to 0$. Therefore, the Richardson model is an exceptional point, owing to the discontinuity in the behaviour of RDM as $\theta \to 0$ and the Richardson model at $\theta \equiv 0$ in the thermodynamic limit $N \to \infty$ [2]. The transition at another special point $\theta = \pi/2$ is continuous as the only level $E_0 = 0$ goes to zero at that value.

In the many-body sector of the Richardson model and RDM, there is one BCS-like ground state, or a whole hierarchy of such states, and the gap(s) from these states to the rest becomes extensive at $\gamma < 2$. The single-particle sector of these models demonstrates the same structure of gapped or energy stratified levels, and moreover, the number of such levels also scales similarly with the system size $N$. In the Richardson model (and other long-range fully correlated models [26–28]) as well as in RDM, the energy stratified levels are special: they form a measure zero subset of all the spectral states, but give the main contribution to the hopping term $j_{mn}$. The most high-energetic of these states are barely affected by the disorder term, and thus, stay non-ergodic in the momentum basis due to their extensive diagonal energy. This leads to both, the ergodicity of these states in the real space, and the fact that they give the main contribution to the hopping term.

Indeed, the disorder term $\varepsilon_n$ (3) in the momentum-space basis (6) plays the role of scattering between plane waves (or hopping) with translation-invariant Gaussian i.i.d. amplitudes $J_{p-q} = \frac{1}{N}\sum_n e^{\frac{2\pi i n(p-q)}{N}} \varepsilon_n$, with zero mean and the variance scaled down with the system size

$$\langle J_p \rangle = 0, \quad \langle J_p^2 \rangle = \frac{W^2}{N} \ . \tag{9}$$

Thus, the corresponding representation of RDM in the momentum-space is a translation-invariant realization of the Rosenzweig-Porter ensemble with a special choice $E_p$ of the diagonal disorder. For this model introduced in [15] it is known that the Fermi's Golden rule is applicable and gives the following broadening

$$\Gamma_p = \frac{2\pi}{\hbar}\rho(E_p)\sum_p |J_{p-q}|^2 \sim \rho(E_p)W^2 \tag{10}$$

of the Breit-Wigner approximation for the eigenstate (see, e.g., [19,32,33])

$$|\psi_{E_p}(p')|^2 \sim \frac{C}{(E_p - E_{p'})^2 + \Gamma_p^2} \ . \tag{11}$$

Here, $C$ is an unimportant normalization constant and we labelled the high-energy eigenstates with disorder-free energy $E_p$ assuming smallness of the broadening $\Gamma_p$ with respect to it. One should note that, unlike the Rosenzweig-Porter model, the RDM in the momentum space has a highly inhomogeneous density of states (DOS) $\rho(E_p)$, $p = 2k+1$, given by

$$\rho(E_p) \simeq \left|\frac{dp}{dE_p}\right| \sim \min\left(\frac{\pi p^2}{4\sin\theta N^{1-\gamma/2}}, \frac{1}{W}\right) \ , \tag{12}$$

---

[2]The similar discontinuous character of the limit is known for the Richardson model in a different class of deformations [15,27], related to the power-law decaying hopping term $j_{mn} = 1/d(m,n)^a$ called the Burin-Maksimov model [26]

where we have taken into account that the disorder $\varepsilon_n \sim W$ hybridizes the levels as soon as the disorder-free version of DOS $|dp/dE_p|$ goes above its bare disorder counterpart $|dn/d\varepsilon_n| \sim 1/W$.

The support set $\Delta p$ occupied by the eigenstate (11) in the momentum space can be found using the condition

$$|E_{p+\Delta p} - E_p| \simeq \Gamma_p \,, \tag{13}$$

which implies non-ergodic behavior as soon as $\Delta p \propto N^{D(p)}$ scales as a fractional $D(p) < 1$ power of $N$.

As soon as $|E_{p+\Delta p} - E_p| \simeq |E_p|$ (or $\Delta p \simeq p$), the condition (13), using (8), (10), and (12), leads to the number $p^*$ of energy-stratified states which are non-ergodic in the momentum basis:

$$p^* \simeq \frac{2}{W} \frac{\sin\theta}{\pi} N^{1-\gamma/2} \,, \quad \Gamma_{p^*} \simeq W \,. \tag{14}$$

The energies of these states are barely affected by disorder as $E_p$ are extensive, $E_p \gg W$ for $|p| \ll p^*$. Thus, our criterion is consistent with the so-called Mott's principle (see, e.g., [15]), which claims that as soon as the bare diagonal energy $E_p$ of a state is large compared to the spectral width $W$ of the hopping term $J_{p-q}$, this state is non-ergodic in the corresponding (momentum) basis. Note that for $\theta < \theta_c \sim N^{-(1-\gamma/2)}$ only the state $p = 0$ is non-ergodic (localized) in the momentum space. Note also that the support set $\Delta p$ is limited from above by $p < p^* \sim N^{1-\gamma/2}$. Thus, from the condition $D(p) < 1$, we see that Eq. (14) is valid until $p^* \ll N$, i.e. for $\gamma > 0$. Henceforth, we will mostly focus our considerations to this parameter interval, $0 < \gamma < 2$.

# 4 Large energy RG in the momentum space & Effective Hamiltonian

In the paper [7], the authors consider renormalization over the matrix size $N$ in the disorder-free RDM with linearly increasing diagonal terms $\varepsilon_n \sim n$. Each RG step involves the removal of one row and one column corresponding to the largest diagonal element $\varepsilon_N$. In particular, the RG considered in [7] can be described as follows:

1. Start with the matrix of size $N_0$ and reduce its size by one at each step.

2. For this, at each step, take the largest absolute diagonal energy ($\varepsilon_N$ or $\varepsilon_1$), and assuming it to be large with respect to the rest of the levels and the hopping terms,

$$|\varepsilon_N| \gg j_{Nn} \,, \tag{15}$$

resolve the eigenproblem with respect to the site $i = N$ corresponding to the level $\varepsilon_N$:

$$(\varepsilon_m - E)\,\psi_E(m) - \sum_n j_{mn}\psi_E(n) = 0 \tag{16a}$$

$$\psi_E(N) = \frac{\sum_{n \neq N} j_{Nn}\psi_E(n)}{\varepsilon_N + j_{NN} - E} \tag{16b}$$

$$(\varepsilon_m - E)\,\psi_E(m) - \sum_{n \neq N} j_{mn}(1)\psi_E(n) = 0 \,, \tag{16c}$$

where $m \neq N$ and $j_{mn}(1)$ is calculated by one RG step

$$j_{mn}(r+1) = j_{mn}(r) + \frac{j_{mN}(r)j_{Nn}(r)}{\varepsilon_N + j_{NN}(r) - E} \,, \tag{17}$$

with $j_{mn}(0) = j_{mn}$.

3. Next, assume $\varepsilon_N + j_{NN} - E \sim W$ and using the ratio $W/\delta = N$ one obtains cyclic RG equations.

In the disordered RDM, the procedure discussed above fails as the maximal diagonal matrix-element does not correspond to the maximal (or minimal) index, which breaks down the self-similar structure of the matrix at further RG steps (see, e.g., [34]). However, one can consider a renormalization group analogous to Eq. (17) by taking into account the large diagonal terms in the *momentum* basis, where the diagonal energies $E_p$ (7) and the hopping terms $J_{p-q}$ (9) satisfy the inequality (15), since

$$|E_p| \gg |J_{p-q}| \ . \tag{18}$$

The corresponding equation for the hopping term at $r$th step of the renormalization, for removal of the level with momentum $p_r$, is given by

$$J_{p,q}(r+1) = J_{p,q}(r) + \frac{J_{p,p_r}(r)J_{p_r,q}(r)}{E_{p_r} + E - J_{p_r,p_r}(r)} \ , \tag{19a}$$

$$J_{p,q}(0) = J_{p-q} \ , \tag{19b}$$

while the diagonal terms stay the same

$$E_q(r) = E_q \text{ for } q \neq p_1, \ldots, p_r \ . \tag{20}$$

Of course, the described renormalization works only when Eq. (18) is valid, i.e. at least for $\gamma < 3$, which is trivially satisfied in our interval of the interest, $0 < \gamma < 2$.

Further, the removal of the level with the largest $|E_p|$ proceeds in the following order, for $s$ up to $s = r \leq N/4$

$$p_0 = 0, \quad p_{2s-1} = -(2s-1), \quad p_{2s} = 2s - 1 \ . \tag{21}$$

Here we should warn the reader that both $\theta = 0$ and $\theta = \pi/2$ have been considered slightly differently. In the vicinity of the Richardson model, $\theta \lesssim N^{-(1-\gamma/2)}$, the only level satisfying Eq. (18) is $E_0$, thus we can consider only one renormalization step $r = 1$. On the contrary, in the vicinity of $\theta = \pi/2$, the level $E_0$ invalidates (18), thus we should start with $s = 1$. However, as we will see in Eqs. (27) and (31), the latter choice does not change the results.

Further, we plan to find the optimal number of RG steps needed for writing the effective Hamiltonian for the bulk spectral states, $E \sim O(1)$, with suppressed correlations. The effects of the energy-stratified states will be taken into account by the RG. In order to find the effective Hamiltonian, in subsection 4.1, we first simplify the RG flow (19) focusing on the leading contributions by order of magnitude. Next, in subsection 4.2, we rewrite the Hamiltonian in the coordinate basis in order to find the optimal number $r$ of RG steps needed to minimize the broadening $\Gamma$, found by Fermi's Golden rule from the effective Hamiltonian.

## 4.1 Simplification of RG (19)

In order to simplify the RG equation (19), here we show that its main contribution is given by

$$\bar{J}_{p,q}(r+1) = J_{p-q} + S_{p,q}(r) \ , \tag{22}$$

where

$$S_{p,q}(r) = \sum_{k=0}^{r} \frac{J_{p-p_k} J_{p_k-q}}{E_{p_k} + E - J_{p_k-p_k}} \ . \tag{23}$$

Indeed, this takes into account the renormalization (19) itself, but neglects the renormalization of the hopping terms $J_{p,q}(r)$ in the sum $S_{p,q}(r)$. As we show in Appendix A, for $r$ smaller than

$$r \ll r^{**} = N^{1-\gamma/3} \ , \tag{24}$$

the above approximation works well, leading to $|S_{p,q}(2r)| \ll |J_{p-q}|$, and the difference between $J_{p,q}$ and $\bar{J}_{p,q}$ is at most of the order $|S_{p,q}|$.

Note that the value $r^{**}$, corresponding to momentum $p^{**} = 2r^{**} - 1 \gg p^*$ according to Eq. (21), is large compared to the number $p^*$ of energy-stratified levels (14) for $\gamma > 0$. Thus, one can take into account all the high-energy states within the above RG flow.

## 4.2   Effective model in the coordinate basis.

Now we are in a position to rewrite the effective renormalized model (20) and (22) in the coordinate basis in order to estimate the fractal dimension of the eigenstates. For this purpose, we separate our renormalized Hamiltonian into four terms

$$\begin{aligned}
H_{p,q}(2r) &= J_{p-q} + \frac{J_p J_{-q}}{E_0} + \sum_{l=1}^{r} \frac{a_{p,q,l}}{E_{2l-1}} + E_q \delta_{p,q} \equiv \\
&\equiv H_{p,q}^{(1)} + H_{p,q}^{(2)} + H_{p,q}^{(3)} + H_{p,q}^{(4)} \ ,
\end{aligned} \tag{25}$$

where $a_{p,q,l} = J_{p-2l+1} J_{2l-1-q} - J_{p+2l-1} J_{-2l+1-q}$ and $p, q \neq p_s$, with $0 \leq s \leq r$, and $p_s$ are from Eq. (21). The discrete Fourier transform of the above terms takes the form

$$H_{m,n}^{(k)} = \sum_{p,q \neq \{p_s\}} \frac{e^{\frac{2\pi i(pm-qn)}{N}}}{N} H_{p,q}^{(k)} = \left( \sum_{p,q} + \sum_{p,q=\{p_s\}} - \sum_{\substack{p, \\ q=\{p_s\}}} - \sum_{\substack{p=\{p_s\}, \\ q}} \right) e^{\frac{2\pi i(pm-qn)}{N}} H_{p,q}^{(k)} \ , \tag{26}$$

where we have replaced the summation over $p, q \neq \{p_s\}$ by complementary sums over the whole interval and over $p_s$ in either or both variables. The first summation is given simply by the initial (not truncated) Fourier transform. After some straightforward algebra and neglecting subleading corrections (both given in Appendix B), one obtains the following renormalized Hamiltonian in the coordinate basis at $2r$th step of the RG flow, with $1 \leq r \leq N/4$ and an unimportant constant $c$

$$H_{m,n}(2r) \sim \varepsilon_m \delta_{mn} + \frac{\varepsilon_m \varepsilon_n}{N^{2-\gamma/2} \cos\theta} +$$
$$+ \begin{cases} \frac{2}{\pi} N^{-\gamma/2} \sin\theta \left( 1 - r\frac{m-n}{N} \right) + \frac{i8\pi^2 \varepsilon_m \varepsilon_n (m-n) r^3}{3N^{3-\gamma/2} \sin\theta} - (\varepsilon_m + \varepsilon_n) \frac{r}{N}, & |m-n| \ll \frac{N}{r} \\ \frac{2}{\pi} N^{-\gamma/2} \sin\theta \frac{c}{r} + \frac{i2\pi \varepsilon_m \varepsilon_n}{N^{2-\gamma/2} \sin\theta} \left( c\frac{N^2 \text{sign}(m-n)}{(m-n)^2} + r \right) - \frac{\varepsilon_m + \varepsilon_n}{2\pi|m-n|}, & |m-n| \gg \frac{N}{r} \end{cases} \ . \tag{27}$$

# 5   Generalization of the matrix-inversion trick for Russian Doll model

Here we present an alternative way to derive the effective Hamiltonian of RDM in the momentum space, analogous to Eq. (25), which is free from the approximations of the

above RG. For this purpose, we generalize the matrix-inversion trick developed in [15]. The main idea of the matrix-inversion trick is as follows: given a hopping matrix with large eigenvalues $E_p$, one adds to it, the identity matrix multiplied by a certain constant $E_0$, and inverts this matrix as follows

$$
E|\psi_E\rangle = \left( \sum_p E_p |p\rangle\langle p| + \sum_n \varepsilon_n |n\rangle\langle n| \right) |\psi_E\rangle \Leftrightarrow
$$

$$
\sum_n (E + E_0 - \varepsilon_n)|n\rangle\langle n|\psi_E\rangle = \sum_p (E_p + E_0)|p\rangle\langle p|\psi_E\rangle \Leftrightarrow
$$

$$
\sum_p \frac{1}{E_p + E_0}|p\rangle\langle p| \sum_n (E + E_0 - \varepsilon_n)|n\rangle\langle n|\psi_E\rangle = |\psi_E\rangle . \quad (28)
$$

In this way, the large energies $E_p$ of the hopping matrix, that provide the dominant contribution to the hopping, can be sent to the denominator without changing the basis. Thus, the effective model can be treated with perturbation theory as soon as the parameter $E_0$ is chosen so as to avoid any resonances $E_p + E_0 \gtrsim O(1)$.

In models with one-sided unbounded growth of the spectrum $E_p$ (like the Burin-Maksimov model [15] where $E_{p<p^*} \gg 1$ are large and positive), one can avoid having singularities in the denominator, $E_p + E_0$, by choosing $E_0 < -\min_p E_p \sim O(1)$, and demonstrate wave-function localization. However, in the RDM, the spectrum is unbounded on both sides (see Eq. (8) for positive and negative $k \ll N$) and does not have finite gaps at finite energies in the thermodynamic limit. Thus, one cannot find a suitable $E_0 \sim O(1)$ to avoid the divergence arising from the inversion of $E_p + E_0$ terms.

In order to obtain convergent terms while applying the matrix-inversion trick, one has to invert only *a part* of the spectrum given by high-energy levels $E_{p<p_r}$

$$
E|\psi_E\rangle = \left( \sum_p E_p |p\rangle\langle p| + \sum_n \varepsilon_n |n\rangle\langle n| \right) |\psi_E\rangle \Leftrightarrow
$$

$$
\left[ \sum_n \varepsilon_n |n\rangle\langle n| + \sum_{|p|>p_r} E_p |p\rangle\langle p| \right] |\psi_E\rangle = \left[ \sum_{|p|<p_r} (E - E_p)|p\rangle\langle p| + \sum_{|p|>p_r} E|p\rangle\langle p| \right] |\psi_E\rangle \Leftrightarrow
$$

$$
\left[ \sum_{|p|<p_r} \frac{1}{1 - E_p/E}|p\rangle\langle p| + \sum_{|p|>p_r} |p\rangle\langle p| \right] \left[ \sum_n \varepsilon_n |n\rangle\langle n| + \sum_{|p|>p_r} E_p |p\rangle\langle p| \right] |\psi_E\rangle = E|\psi_E\rangle \Leftrightarrow
$$

$$
\left[ \left( \sum_{|p|>p_r} |p\rangle\langle p| + \sum_{|p|<p_r} \frac{E}{E_p}|p\rangle\langle p| \right) \sum_n \varepsilon_n |n\rangle\langle n| + \sum_{|p|>p_r} E_p |p\rangle\langle p| \right] |\psi_E\rangle = E|\psi_E\rangle , \quad (29)
$$

where, for simplicity, we choose $E_0 = -E$ and use $|E_{|p|<p_r}| \gg E$.

Observe that Eq. (29) corresponds term-by-term with Eq. (25). Indeed,

- the first term is equivalent to $H^{(1)}$ after neglecting subleading terms like $i_{mn}$ in (65) of Appendix B,

- the part of the second term with $p = p_0 = 0$ corresponds to $EH_{mn}^{(2)}/\varepsilon_m$ after neglecting $g_{m,0}\sqrt{r/N}$ subleading terms in (25),

- the rest part of the second term with $p = p_s \neq 0$ corresponds to $EH_{mn}^{(3)}/\varepsilon_m$ after neglecting $g_{m,0}\sqrt{r/N}$ subleading terms in (25),

- while the last term is just equal to $H^{(4)}$.

To sum up, this result shows that all the approximations performed in the previous section in order to derive the effective Hamiltonian, Eq. (27), either lead to subleading corrections or to prefactors $E/\varepsilon_m \sim O(1)$ of order unity. As the matrix-inversion trick provides merely a different representation of the exact eigenproblem without any approximations, the equivalence between Eqs. (25) and (29) confirms the applicability of the effective Hamiltonian (27) in the whole range of parameters of interest, $0 < \gamma < 2$.

# 6 Results

Now we are ready to calculate the non-ergodic properties of eigenstates based on effective Hamiltonian (27) and Fermi's Golden rule approximation (1). Similar to the matrix-inversion trick [15], the Fermi's Golden rule for each effective Hamiltonian with a certain $r$ gives an upper bound for the fractal dimension via the broadening $\Gamma_n(2r)$ (see definition 30). Therfore, in order to find the true fractal dimension, one should make the upper bound strict by finding the optimal $r = r_{\text{opt}}$ that minimizes $\Gamma_n(2r)$. Below, we implement the optimization procedure analytically, and later verify our result for the fractal dimension using numerical calculations of eigenstate statistics.

## 6.1 Analytical results – Optimization of fractal dimension

Similar to the Rosenzweig-Porter model, we expect the model given by the effective Hamiltonian (27) to exhibit only fractal (defined via level broadening in (33)) and not multifractal states (where multifractal dimensions $D_q$ are parameterized by the order $q$ of the wave-function moment, see Sec. 6.2). Therefore, in order to calculate the fractal dimension $D$ of eigenstates in RDM, we use Fermi's Golden rule analogous to (1), but with the effective hopping term from the renormalized Hamiltonian (27)

$$\Gamma_n(2r) = \frac{2\pi}{\hbar}\rho(E_n) \sum_{m \neq n} |H_{m,n}(2r)|^2 \ . \tag{30}$$

Taking the energies and DOS in the bulk of the spectrum, $E_n \sim \varepsilon_n \sim W$ and $\rho(E_n) \sim 1/W$, one obtains, for each term of the effective Hamiltonian, the following expression [3]

$$\frac{W}{2\pi}\Gamma_n(2r) \sim \frac{W^4}{N^{3-\gamma}\cos^2\theta} + \frac{4}{\pi^2}N^{-\gamma}\sin^2\theta\left[\sum_{n=0}^{N/r}\left(1 - \frac{r}{N}n\right)^2 + \frac{c^2}{r^2}\left(N - \frac{N}{r}\right)\right] +$$

$$+\frac{W^4}{N^{4-\gamma}\sin^2\theta}\left[\sum_{n=0}^{N/r}n^2\frac{64\pi^4r^6}{9N^2} + \sum_{n=N/r}^{N}\frac{c^2N^4}{16\pi^2n^4} + r^2\left(N - \frac{N}{r}\right)\right] + W^2\left[\frac{r^2}{N^2}\frac{N}{r} + \sum_{n=N/r}^{N}\frac{1}{4\pi^2n^2}\right] \sim$$

$$\sim \frac{W^4}{N^{3-\gamma}\cos^2\theta} + \frac{4}{\pi^2}\sin^2\theta\frac{N^{1-\gamma}}{r} + \frac{W^4r^3}{N^{3-\gamma}\sin^2\theta} + \frac{W^2r}{N} \tag{31}$$

The first term (corresponding to $H^{(3)}$ with $p = p_0$) is subleading for all $|\theta| \gg N^{-(1-\gamma/2)}$ and $r \gg 1$. Formally, in the vicinity of $\theta = \pi/2$ this term diverges, but as we discussed earlier, the inequality (18) should be satisfied in order to write this term. For similar reasons, in the vicinity of $\theta = 0$, the divergence of the third term $\sim 1/\sin\theta$ can be ignored.

---

[3]Here we neglect the cross-terms as we are interested in the dominant contributions and the competition between them at different $r$.

After neglecting the first term for finite $\theta$, the rest of the three terms give the optimal value of $r$ corresponding to the minimal level broadening

$$r_{\text{opt}} \simeq c \frac{\sin\theta}{W} N^{1-\gamma/2} \;\Leftrightarrow\; \Gamma_n(r_{\text{opt}}) \simeq c' \sin\theta N^{-\gamma/2} \;, \tag{32}$$

with certain constants $c$ and $c'$ of order one. Note that this optimal value $r_{\text{opt}}$ satisfies the condition (24), $r \ll r^* \sim N^{1-\gamma/3}$, for all $\gamma > 0$. The validity condition $|J_{p,q}(r) - \bar{J}_{p,q}(r)| \ll |S_{p,q}(r)|$ for the above used RG, leading to $r \ll N^{(3-\gamma)/4}$ from Eq. (56) in Appendix A, is satisfied for $\gamma > 1$. However, even for $r \gtrsim N^{(3-\gamma)/4}$, corresponding to $0 < \gamma < 1$, $J_{p,q}(r) - \bar{J}_{p,q}(r)$ gives at most the same-order contribution as $S_{p,q}(r)$ and affects only the numerical prefactors $c$ and $c'$.

In the bulk of the spectrum, as the mean level spacing is $\delta = 1/(\rho(E)N) \sim W/N$, the fractal dimension for the typical wave function in this case should be given by the ratio

$$N^D = \Gamma_n(r_{\text{opt}})/\delta \sim \frac{\sin\theta}{W} N^{1-\gamma/2} \;\Leftrightarrow\; D = 1 - \gamma/2 \;, \tag{33}$$

which is different from the one in the Rosenzweig-Porter model [13]. This result is also confirmed by numerical calculations below, where we define the fractal dimension via the inverse participation ratio (but not as the number of sites where the wave function has significantly non-zero values within the Breit-Wigner approximation). Note that in cases where the Fermi's Golden rule does not work for the initial problem, the number $p^*$ of the energy-stratified levels, Eq. (14), determines the fractal dimension via the expression $p^* \sim N^D$ (see [17] for more details).

The effective renormalized Hamiltonian (27) in this case is given by

$$H_{m,n}(2r_{\text{opt}}) \simeq \varepsilon_m \delta_{mn} - \begin{cases} \left(\frac{\varepsilon_m + \varepsilon_n}{W} - c\right)\Gamma, & |m-n| \ll W/\Gamma \\ \frac{\varepsilon_m + \varepsilon_n}{2\pi|m-n|}, & |m-n| \gg W/\Gamma \end{cases} , \tag{34}$$

where $W/\Gamma \simeq N/r_{\text{opt}} \sim (W/\sin\theta) N^{\gamma/2}$ and we neglect the subleading terms for simplicity.

Note that the Hamiltonian (34) is equivalent to the one of power-law random banded matrices [35] with a bandwidth $b = W/\Gamma$ and diagonal disorder $W$ rescaled as $\sim N^{\gamma/2}$.

- if none of $b$ and $W$ scales with $N$, the system hosts genuinely *multifractal* states, with $D_q \sim b$ at $b \ll 1$ and $1 - D_q \sim 1/b$ at $b \gg 1$ [35],

- the scaling $b \sim N^{\gamma/2}$, $W = O(1)$ sends the system into the ergodic phase, with $D = 1$,

- the scaling $W \sim N^{\gamma/2}$, $b = O(1)$ leads to the localization, $D = 0$,

- whereas the case of RDM, corresponding to the simultaneous scaling of both parameters $W \sim b \sim N^{\gamma/2}$, gives fractal eigenstates described by the Fermi's Golden rule (30) [4].

The bulk eigenstates of such an effective Hamiltonian should be given by two contributions:

- First, due to the presence of Rosenzweig-Porter-like long-range hopping terms at $|m-n| \ll W/\Gamma$, the wave-function should have a contribution of a Lorentzian profile versus $\varepsilon_n$ with the width $|E_m - \varepsilon_n| \sim \Gamma$ [19,32,33]

$$|\psi_{E_m}(n)|^2 \simeq \frac{\delta\Gamma}{(E_m - \varepsilon_n)^2 + \Gamma^2} \;. \tag{35}$$

---

[4]However, there are studies which show that in such a case the system might have eigenstates, being spatially multifractal, while the local density of states has an additional fractal structure in the energy spectrum inside a miniband of size $\Gamma \sim N^{-\gamma/2}$ provided by the Fermi's Golden rule [24].

- Second, similar to the power-law banded random matrices $\langle |j_{mn}|^2 \rangle \sim |m - n|^{-2a}$ at the critical power $a = 1$, there should be the multifractal proliferation of the wave-function maxima given by the resonances. However, unlike the power-law banded random matrix case, in the RDM, the $N$-scaling of the cutoff $W/\Gamma \sim N^{\gamma/2}$, at which this power-law comes into play, significantly reduces the number of resonances:

$$N_{res} \sim \sum_m |H_{mn}|/W \sim \ln\left(N\Gamma/W\right) = (1 - \gamma/2)\ln N \tag{36}$$

and does not affect the fractal dimension of the system, given by (see e.g., Eqs. (30-31) in [36])

$$D \ln N \sim N_{res} \iff D = 1 - \gamma/2 . \tag{37}$$

## 6.2 Numerical results – Spectrum of fractal dimensions, level statistics, and wave-function decay

In numerics, we proceed free of approximations and consider exact diagonalization of the initial model (2), calculating eigenstates $\psi_{E_n}(m)$ in the coordinate basis and eigenvalues $E_n$. We analyze the data using probes of multifractality while focusing on the mid-spectrum states. For this purpose, we consider two relevant measures of eigenfunction statistics based on the distribution of amplitudes $P(|\psi_E(n)|^2)$.

First, we look at the *spectrum of fractal dimensions*, defined as the power $f(\alpha)$ of the scaling of the distribution, $P(\alpha) \sim N^{f(\alpha)-1}$ of $\alpha = -\ln|\psi_E(n)|^2/\ln N$ [37],

$$f(\alpha) = 1 - \alpha + \lim_{N \to \infty} \frac{\ln[P(|\psi_E(n)|^2 = N^{-\alpha})]}{\ln N} . \tag{38}$$

The spectrum $f(\alpha)$ is a kind of large deviation function showing the tails of the distribution of $\ln|\psi_E(n)|^2$ far from its typical (most probable) value

$$\left\langle \ln|\psi_E(n)|^2 \right\rangle = -\alpha_0 \ln N . \tag{39}$$

It has a bunch of properties (normalization condition of the probability distribution $f(\alpha) \leq 1$, $f(\alpha_0) = 1$, or the wave-function normalization $f(\alpha) \leq \alpha$, $f(\alpha_1) = \alpha_1$), among which we specifically mention the symmetry [37], originally discovered in [38],

$$f(2 - \alpha) = f(\alpha) - (\alpha - 1) , \tag{40}$$

relating peaks (small $\alpha < 1$) and tails (large $\alpha > 1$) of the wave-function. This symmetry is known to work for non-ergodic extended states, fractal or multifractal. In particular, for the Rosenzweig-Porter model, as shown in [13], $f(\alpha)$ takes a simple linear form for $\gamma \geq 1$, with an additional point $f(0) = 0$ for $\gamma > 2$:

$$f_{RP}(\alpha) = \begin{cases} 1 + (\alpha - \gamma)/2, & \max(0, 2 - \gamma) < \alpha < \gamma \\ -\infty, & \text{otherwise} \end{cases} \tag{41}$$

The above spectrum satisfies the symmetry (40) for $\gamma < 2$.

The second widely used probe of multifractality that we consider is the inverse participation ratio $I_q$ defined via the moments of eigenstates, as follows,

$$I_q = \sum_n |\psi_E(n)|^{2q} \sim N^{-(q-1)D_q} , \tag{42}$$

where the $q$-dependent exponents $D_q$ are called fractal dimensions. In the ergodic phase, $D_q = 1$ for all $q$, whereas in the case of localization, $D_q = 0$ for $q > 0$. The intermediate

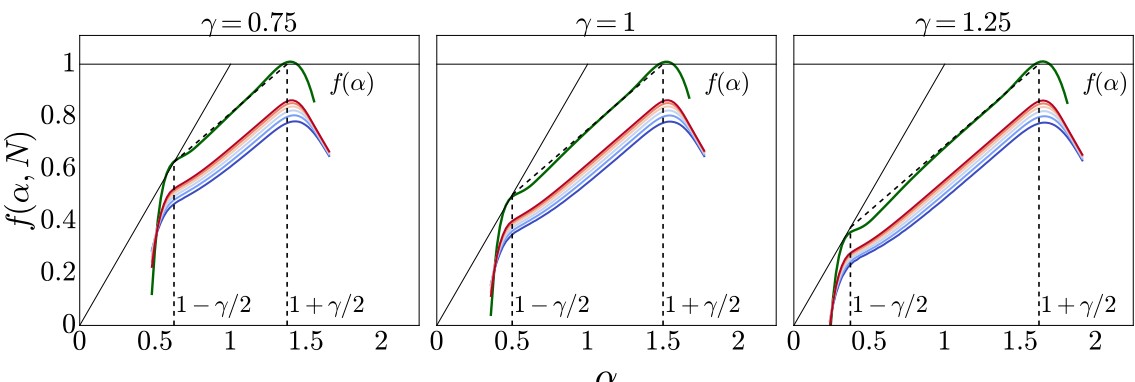

Figure 2: **The spectrum of fractal dimensions** $f(\alpha)$ in RDM model for 50 % mid-spectrum eigenstates, $\theta = 0.25$, and (left) $\gamma = 0.75$, (middle) $\gamma = 1$, and (right) $\gamma = 1.25$. $f(\alpha)$ is extrapolated (green) from $N = 2^9 - 2^{14}$ (from blue to red) with 1000 disorder realizations for each. The dashed line shows the analytical prediction (46).

values, $0 < D_q < 1$, correspond to non-ergodic extended states, which can be either *multifractal*, where $D_q$ is represented by a strictly decaying function of $q$, or *fractal*, where $D_q = D$ does not depend on $q$, at least for $q > 1/2$.

The relation between $D_q$ and $f(\alpha)$ is given by the saddle-point approximation for disorder-averaged moments,

$$\langle I_q \rangle = N \int |\psi|^{2q} P(\psi) d\psi = \int_0 N^{f(\alpha)-q\alpha} d\alpha \simeq N^{\max_\alpha [f(\alpha)-q\alpha]} \,, \tag{43}$$

and the Legendre transform:

$$(q-1)D_q = q\alpha_q - f(\alpha_q), \quad \text{where } f'(\alpha_q) = q \,, \tag{44}$$

where the last equality is the definition of $\alpha_q$.

From the above definition, one can immediately see that $\alpha_1$, determining the wave-function normalization condition $f(\alpha_1) = \alpha_1$, gives the limiting value $D_{q \to 1}$

$$D_1 = \alpha_1 = 2 - \alpha_0 \,, \tag{45}$$

where the latter equality is given by the symmetry (40). Here $\alpha_0$ determines the most typical wave-function amplitudes (39) and consequently, gives the maxima, $f(\alpha_0) = 1$.

Using the above relation (45) and our prediction for value of the fractal dimension, Eq. (33), one can find the spectrum of fractal dimension in the RDM, similar to (41),

$$f(\alpha) = \begin{cases} 1 + \frac{(2\alpha - 2 - \gamma)}{4}, & \max\left(0, 1 - \frac{\gamma}{2}\right) < \alpha < 1 + \frac{\gamma}{2} \\ -\infty, & \text{otherwise} \end{cases} \,. \tag{46}$$

Alternatively, one can derive the above expression from the Breit-Wigner approximation (11) using the broadening (33), see e.g. [33].

Numerically, as one cannot achieve infinite system sizes, both the spectrum of fractal dimensions Eq. (38) and the fractal dimensions themselves Eq. (42), are calculated for different finite system sizes and extrapolated to the thermodynamic limit $N \to \infty$ (see Appendix C). The spectrum of fractal dimension at finite system sizes can be extracted directly from the histogram over $\alpha$ (see, e.g., [13,15,39,40]), while the inverse participation ratio is given simply by Eq. (42).

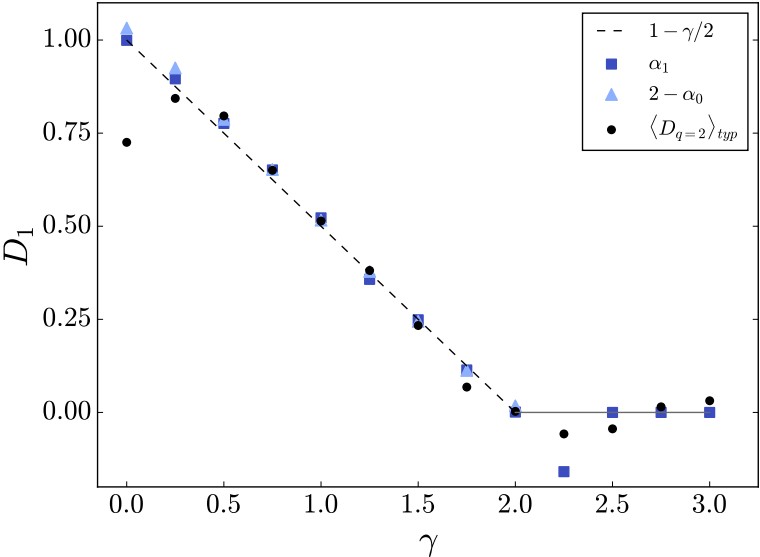

Figure 3: **The fractal dimension $D_1$ versus $\gamma$** in RDM model for 50 % mid-spectrum eigenstates, and $\theta = 0.25$. The symbols correspond to $D_q$ extracted from the inverse participation ratio ($\bullet$) and from the points corresponding to the first $\alpha_1$ ($\square$), and zeroth $2 - \alpha_0$ ($\triangle$) moments of $P(\alpha) \sim N^{f(\alpha)-1}$. The black dashed line shows the analytical prediction (33). The symmetry (40) used for $2 - \alpha_0$ works only for the delocalized phases, i.e. we plot $2 - \alpha_0$ only for $\gamma \leq 2$. The data is extrapolated from $N = 2^9 - 2^{14}$ with 1000 disorder realizations for each.

Figure 2 shows the spectrum of fractal dimensions $f(\alpha)$ extracted from the numerical simulations. One can see that the extrapolation procedure gives the correct normalization value $f(\alpha_0) = 1$, and good agreement with the analytical prediction (46).

One can use either the inverse participation ratio (42), or its relation to $f(\alpha)$ (45), in order to extract the fractal dimension, see Fig. 3. Agreement amongst these two measures and with the analytical prediction (33) confirms our analytical derivation and the symmetry (40) of $f(\alpha)$. Slight deviations from the prediction in the localized phase are caused by finite-size effects, which are enhanced close to the Anderson transition.

Note that, according to the analytical prediction, the fractal dimension is not homogeneous across the spectrum and this is also confirmed numerically, see Fig. 4: The fractal dimension in the spectral bulk (more than 90 % of states at the considered system sizes) shows the above fractal behavior for $0 < \gamma < 2$, while at the edges, the high-energy states become ergodic with $D_2 \to 1$.

In addition to the previous two measures, we also consider the eigenvalue statistics using a so-called adjacent level gap ratio (defined in [41,42]) and the wave-function spatial decay (first used in [27]). The ratio statistics considers

$$r_n = \frac{\min(s_n, s_{n+1})}{\max(s_n, s_{n+1})}, \text{ where } s_n = E_{n+1} - E_n . \tag{47}$$

Localization corresponds to Poisson level statistics, characterized by the absence of level repulsion, and leads to mean $r = \langle r_n \rangle = r_P = 2\ln 2 - 1 \simeq 0.3863$. The ergodic random-matrix prediction corresponds to the Wigner surmise [6] and is given by $r_{GOE} \simeq 0.5307$, for orthogonal symmetry and $r_{GUE} \simeq 0.5996$ for the unitary one. For the fractal phase of the Rosenzweig-Porter kind, the gap ratio still shows the Wigner-Dyson value in the entire delocalized phase, $\gamma < 2$, while non-ergodicity is visible only at higher order gap ratios (see,

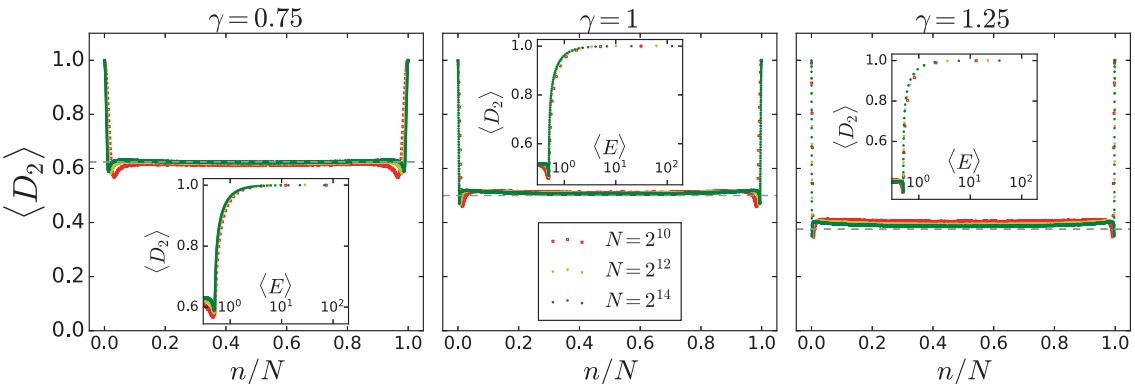

Figure 4: **The fractal dimension $D_2(E_n)$ versus energy index $n$** in RDM model for $\gamma = 0.75$, 1, 1.25, $\theta = 0.25$, and $N = 2^{10}, 2^{12}, 2^{14}$. (insets) the same data for $D_2(E_n)$, shown vs energy $E_n$ and zoomed close to the right spectral edge. The data both for $D_2(E_n)$ and $E_n$ are averaged over 1000 realizations of disorder for each eigenvalue index separately.

e.g., [29, 43, 44]) or in other measures like the spectral form factor, level compressibility [13] or the power spectrum [45–47].

Figure 5 shows the conventional ratio statistics (47) across the spectrum in disordered RDM for $\gamma = 0.75$, 1, 1.25. One can see that, similar to the Rosenzweig-Porter model, the disorder-averaged ratios in the bulk of the spectrum follow the Wigner-Dyson unitary value $r_{GUE}$, while the spectral edges states correspond to special $r$ values, first going up to the equidistant spectral average $r = 1$, and then to the nearly degenerate levels $r \simeq 0$.

Our last measure, the wave-function spatial decay [15, 27, 29, 48], uncovers the structure of the eigenstates predicted by the effective Hamiltonian (34) and Eq. (35). In order to observe the wave-function spatial decay, we plot it versus the diagonal energy differences $|\varepsilon_m - \varepsilon_n|$, provided these energies are ascendingly ordered $\varepsilon_n < \varepsilon_{n+1}$, see Fig. 6.

Indeed, according to (35), for sites $n$ with the diagonal energy $\varepsilon_n$ close to the eigenvalue $E_m$, $|\varepsilon_n - E_m| \lesssim \Gamma$, the wave-function has a fractal structure $|\psi_{E_m}(n)|^2 \sim N^{-D}$ like in the Rosenzweig-Porter model. This is confirmed by the Lorentzian form of the wave-function decay in Fig. 6, similar to the Rosenzweig-Porter results [48].

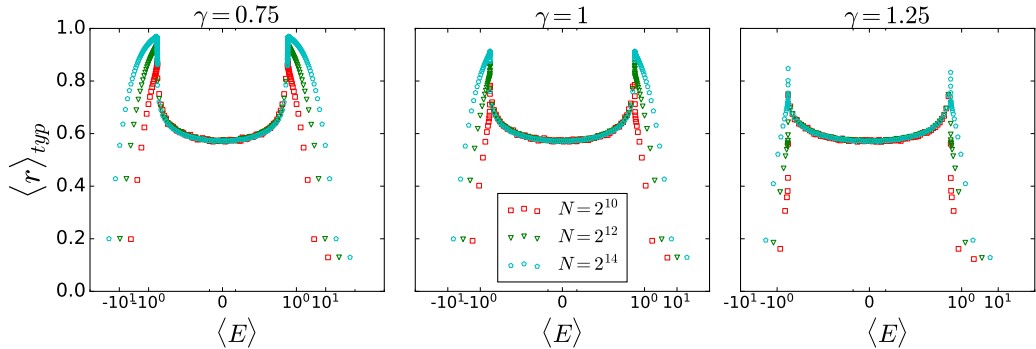

Figure 5: **The spectral ratio statistics $\langle r \rangle_{typ} = \exp \langle \ln r(E_n) \rangle$ versus energy $E_n$** in RDM model for $\gamma = 0.75$, 1, 1.25, $\theta = 0.25$, and $N = 2^{10}, 2^{12}, 2^{14}$. The data both for $r(E_n)$ and $E_n$ are averaged over 1000 realizations of disorder for each eigenvalue index separately.

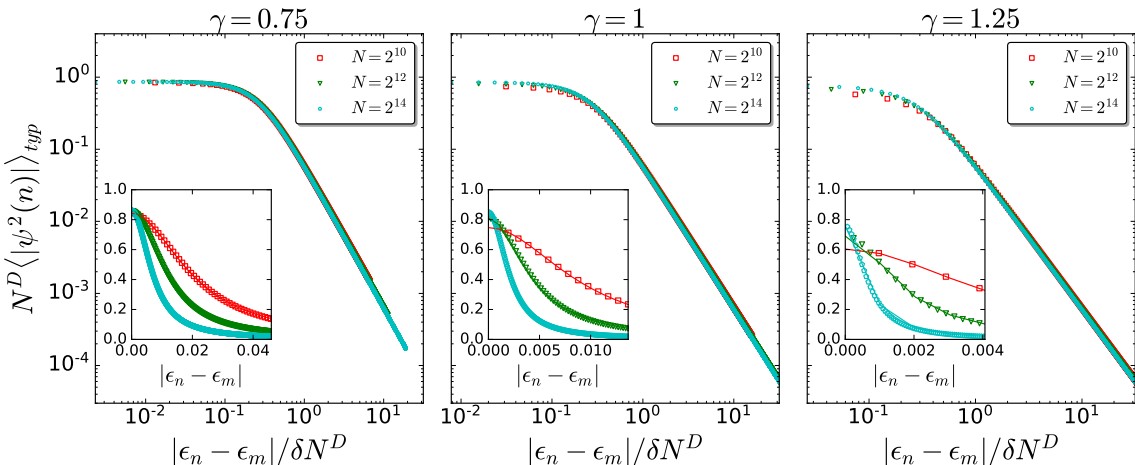

Figure 6: **The wave-function spatial decay** $N^D \left\langle |\psi_{E_m}(n)|^2 \right\rangle$ **versus the diagonal energy differences** $|\varepsilon_n - \varepsilon_m|$ in RDM model for $\gamma = 0.75$, 1, 1.25, $\theta = 0.25$, and $N = 2^{10}, 2^{12}, 2^{14}$. The data is averaged over 1000 realizations of disorder for each eigenvalue index separately. The main plots show the collapse of the curve with the rescaling $\delta \cdot N^D$ of the energy differences in the log-log scale. The insets show the linear scale without $x$-axis rescaling.

## 7 Conclusion and Outlook

In summary, we consider the localization and ergodicity-breaking properties of eigenstates in the disordered Russian Doll model, with generalized amplitude of the coupling strength $\sim N^{-\gamma/2}$. In this regard, we develop RG flow based on the renormalization of high-energy states in the momentum basis, and provide an effective Hamiltonian, valid for any number of renormalized high-energy states within the considered parameter range.

In addition, we validate our result for the effective Hamiltonian and confirm that the approximations we used are valid to leading order in the parameter $r/r^{**}$, i.e. the number of the RG steps normalized by the sub-extensive upper bound $r^{**} \sim N^{1-\gamma/3}$, and all subsequent corrections are subleading. We do so by generalizing the matrix-inversion trick to cases where the hopping-spectrum is dense at all finite energies and grows without bound on both sides of the spectrum. Unlike the localized case, described by the matrix-inversion, the above spectral structure corresponds to the phases of *delocalized* eigenstates that can be understood using the above generalization of the matrix-inversion trick (cf. [17]).

Based on effective Hamiltonian, we find the fractal dimension of eigenstates in the bulk of the spectrum. We see that RDM has a delocalised non-ergodic phase for an extended range of parameter values ($0 < \gamma < 2$) compared to the Rosenzweig-Porter model ($1 < \gamma < 2$), and with a different fractal dimension $D = 1 - \gamma/2$.

Note that, as the fractal properties in RDM barely depend on the time-reversal symmetry breaking parameter $\theta \neq 0$, the Richardson model provides an example of an exceptional point, since the limiting behavior $\theta \to 0$ of the Russian Doll model does not correspond to that of the Richardson model, $\theta = 0$. Unlike the Burin-Maksimov model [15, 26, 27] – where the symmetry-breaking parameter destroys the BA integrability, and thus, one expects to see discontinuous behaviour – this work demonstrates that broken time-reversal symmetry in RDM, which preserves the BA integrability for any finite $N$, may lead to a similar discontinuity.

It would be interesting to look for non-ergodicity in the ensemble of twisted XXX

models with random inhomogeneities, using their relation to the RDM model. Another interesting direction for further study concerns the derivation of the generalized RDM, that is, Richardson model with broken time-reversal symmetry and the generalized hopping term scaling, and the investigation of its fractal properties. It is an important direction of research as the generalized Richardson model describes the superconducting phase of the mixture of Sachdev-Ye-Kitaev and Fermi-Hubbard models [49].

Moreover, we show that the Russian Doll model provides an example of a Bethe-Ansatz integrable model with delocalized non-ergodic eigenstates already in the single-particle sector. This raises concerns about the relation between Bethe-Ansatz integrability and localization and opens a new avenue in this research direction.

## Acknowledgements

A. G. and I. M. K thank IIP-Natal, where the project has been initiated during the program "Random geometries and multifractality in condensed matter and statistical mechanics" for the hospitality. V. R. M. and I. M. K acknowledge technical support by MPIPKS Dresden.

**Funding information**   I. M. K. acknowledges the support by Russian Science Foundation (Grant No. 21-72-10161).

## A    Estimation of smallness of a parameter $S_{p,q}(2r)$ and $J_{p,q}(r) - \bar{J}_{p,q}(r)$

Within the condition (24), $r \ll r^{**} = N^{1-\gamma/3}$,

$$E_{p_r} \gg J_{p-q} \sim N^{-1/2} \tag{48}$$

and the sum $S_{p,q}(r)$, Eq. (23), is small compared to $J_{p-q}$

$$|S_{p,q}(2r)| \leq \left| \sum_{k=0}^{r} \frac{J_{p-p_k} J_{p_k-q}}{E_{p_k}} \right| \simeq \frac{1}{N^{1-\gamma/2}} \left| \frac{1}{N\cos\theta} + \frac{\pi}{2\sin\theta} \sum_{l=1}^{r} (2l-1) a_{p,q,l} \right| , \tag{49}$$

where $a_{p,q,l} = J_{p-2l+1} J_{2l-1-q} - J_{p+2l-1} J_{-2l+1-q}$ due to (9) has zero mean and the following variance

$$\langle a_{p,q,l} \rangle = 0 , \tag{50a}$$

$$\left\langle |a_{p,q,l}|^2 \right\rangle = \frac{2}{N^2}(1 + \delta_{p,q}) . \tag{50b}$$

Using this, the above sum of random variables can be approximated via its variance (as it have the zero mean)

$$\sum_{l=1}^{r} (2l-1)^2 \left\langle |a_{p,q,l}|^2 \right\rangle \simeq \frac{2r(4r^2-1)}{3N^2}(1 + \delta_{p,q}) . \tag{51}$$

Finally, this gives the following estimate for $S_{p,q}(2r)$ at $r \gg 1$

$$|S_{p,q}(2r)| \lesssim \frac{1}{N^{2-\gamma/2}} \left[ \frac{1}{\cos\theta} + \frac{r^{3/2}}{\sin\theta} \right] \ll J_{p-q} \simeq \frac{1}{N^{1/2}} , \tag{52}$$

due to $\gamma < 3$ and (24). In a similar way one can estimate the difference

$$l_{p,q}(r) = J_{p,q}(r) - \bar{J}_{p,q}(r) , \quad l_{p,q}(1) = 0 . \tag{53}$$

Indeed, using Eqs. (19), (22), and (48) one immediately obtains

$$l_{p,q}(r+1) - l_{p,q}(r) \simeq \frac{1}{E_{p_r}} \left[ J_{p-p_k} \left( S_{p_k,q}(r) + l_{p_k,q}(r) \right) + J_{p_k-q} \left( S_{p,p_k}(r) + l_{p,p_k}(r) \right) \right] , \tag{54}$$

where we neglected the quadratic term $\left( S_{p,p_k}(r) + l_{p,p_k}(r) \right) \left( S_{p_k,q}(r) + l_{p_k,q}(r) \right)$ due to its smallness.

Further we estimate by the order of magnitude the parameter $l$ by rewriting the above equation for $l$ in the continuous form and neglecting the difference between variables with different indices

$$\frac{dl(r)}{dr} \simeq \frac{J}{E_{p_r}} \left( S(r) + l(r) \right) . \tag{55}$$

Solving this ordinary differential equation in the variable $x$

$$N^{-(3-\gamma)/2} \leq x = \frac{Jr}{E_{p_r}} \sim \frac{r^2}{N^{(3-\gamma)/2}} \ll N^{(3-\gamma)/6} \tag{56}$$

one obtains

$$l(r) = -S(r) \frac{\int_0^x y^{3/4} e^y dy}{x^{3/4} e^x} . \tag{57}$$

For $x \ll 1$ one can immediately see that $|l(r)| \sim x S(r) \ll S(r)$.

In the opposite limit of $x \gtrsim 1$ one can only bound $|l(r)| \leq S(r)$ using the condition for $y' = (x - y) \geq 0$ in the integrand

$$\left| \frac{l(r)}{S(r)} \right| = \int_0^x \left( 1 - \frac{y'}{x} \right)^{3/4} e^{-y'} dy' \leq 1 . \tag{58}$$

In this case one cannot neglect $l(r)$ with respect to $S(r)$, but can absorb it to $S(r)$ if the numerical prefactors are not important. Thus, in the main text we consider both above cases of $x \ll 1$ and $x \gtrsim 1$.

## B  Derivation of the effective Hamiltonian (25) in the coordinate basis (27)

In Eq. (25) we separate our renormalized Hamiltonian in four terms

$$H_{p,q}^{(1)} = J_{p-q} , \quad H_{p,q}^{(2)} = \frac{J_p J_{-q}}{E_0} , \quad H_{p,q}^{(3)} = \sum_{l=1}^{r} \frac{a_{p,q,l}}{E_{2l-1}} , \quad H_{p,q}^{(4)} = E_q \delta_{p,q} , \tag{59}$$

where we introduce the notation $a_{p,q,l} = J_{p-2l+1} J_{2l-1-q} - J_{p+2l-1} J_{-2l+1-q}$ and $p, q \neq p_s$, with $0 \leq s \leq r$, and $p_s$ are from Eq. (21). The discrete Fourier transform of the above terms takes the form

$$H_{m,n}^{(k)} = \sum_{p,q \neq \{p_s\}} \frac{e^{\frac{2\pi i(pm-qn)}{N}}}{N} H_{p,q}^{(k)} = \left( \sum_{p,q} + \sum_{p,q=\{p_s\}} - \sum_{\substack{p, \\ q=\{p_s\}}} - \sum_{\substack{p=\{p_s\}, \\ q}} \right) \frac{e^{\frac{2\pi i(pm-qn)}{N}}}{N} H_{p,q}^{(k)} , \tag{60}$$

where we replaced the summation over $p, q \neq \{p_s\}$ by the complemented sums over the whole interval and over $p_s$ in either or both variables. The first summation is given just by the initial (not truncated) Fourier transform.

Here we will calculate all these terms one-by-one. The first term written in the above four sums takes the form

$$
H_{m,n}^{(1)} = \sum_{p,q \neq \{p_s\}} \frac{e^{\frac{2\pi i(pm-qn)}{N}}}{N} J_{p-q} \equiv \varepsilon_m \delta_{mn} + i_{mn} - I_{mn} - K_{mn} ,
\tag{61}
$$

where the first term corresponds to the diagonal disorder, the second one is given by

$$
i_{mn} = \sum_{p,q=\{p_s\}} \frac{e^{\frac{2\pi i(pm-qn)}{N}}}{N} J_{p-q} ,
\tag{62}
$$

while the last two terms are symmetric with respect to each other by the Hermitian conjugation

$$
I_{mn} = \sum_{\substack{p, \\ q=\{p_s\}}} \frac{e^{\frac{2\pi i(pm-qn)}{N}}}{N} J_{p-q} = \sum_{\substack{p'=\{p_s\}, \\ q'}} \frac{e^{-\frac{2\pi i(p'n-q'm)}{N}}}{N} J_{q-p} = K_{nm}^*
\tag{63}
$$

with $p' = q$ and $q' = p$ and $J_{-p} = J_p^*$. Let's calculate, first, $I_{mn}$ by shifting the summation over $p$ to $k = p - q$

$$
I_{mn} = \sum_{q=\{p_s\}} \frac{e^{\frac{2\pi iq(m-n)}{N}}}{N} \sum_k e^{\frac{2\pi ikm}{N}} J_k = \varepsilon_m \sum_{q=\{p_s\}} \frac{e^{\frac{2\pi iq(m-n)}{N}}}{N} = \varepsilon_m \left( \frac{1}{N} + \sum_{l=-r}^{r-1} \frac{e^{\frac{2\pi i(2l-1)(m-n)}{N}}}{N} \right) =
$$

$$
= \varepsilon_m \left( \frac{1}{N} + \frac{\sin\left(4\pi r(m-n)/N\right)}{N \sin\left(2\pi(m-n)/N\right)} \right) , \quad (64)
$$

The second sum $i_{mn}$ in Eq. (61) can be found after the same shift

$$
i_{mn} = \sum_{q=\{p_s\}} \frac{e^{\frac{2\pi iq(m-n)}{N}}}{N} \sum_{k+q=\{p_s\}} e^{\frac{2\pi ikm}{N}} J_k =
$$

$$
= \sum_{q=\{p_s\}} \frac{e^{\frac{2\pi iq(m-n)}{N}}}{N} g_{m,q} \sqrt{\frac{r}{N}} \lesssim \left( \frac{1}{N} + \frac{\sin\left(4\pi r(m-n)/N\right)}{N \sin\left(2\pi(m-n)/N\right)} \right) g_{m,q} \sqrt{\frac{r}{N}} \ll I_{mn} \quad (65)
$$

and estimating the following sum with the random phase approximation

$$
\sum_{k=\{p_s-q\}} e^{\frac{2\pi ikm}{N}} J_k \simeq g_{m,q} \sqrt{\frac{r}{N}}
\tag{66}
$$

and the central limit theorem for $r \gg 1$ leading to random variable $g_m$ of the order of one.

After neglecting of the small terms $g_m \sqrt{r/N}$ with respect to $\varepsilon_m$ we obtain for the first term

$$
H_{m,n}^{(1)}(2r) \simeq \varepsilon_m \delta_{mn} - (\varepsilon_m + \varepsilon_n) \frac{\sin\left(4\pi r(m-n)/N\right)}{N \sin\left(2\pi(m-n)/N\right)} \sim
$$

$$
\sim \varepsilon_m \delta_{mn} - (\varepsilon_m + \varepsilon_n) \begin{cases} \frac{r}{N}, & |m-n| \leq \frac{N}{r} \\ \frac{\sin(4\pi r(m-n)/N)}{2\pi|m-n|}, & |m-n| \geq \frac{N}{r} \end{cases} .
\tag{67}
$$

In the last equality we approximate the sine factors of the last term by linear functions when their arguments are small compared to one.

The second term $H_{m,n}^{(2)}$ splits in the product of two equivalent sums, giving in the same approximation as for $i_{mn}$

$$\left(N^{2-\gamma/2}\cos\theta\right)H_{m,n}^{(2)} = \sum_{p,q\neq\{p_s\}} e^{\frac{2\pi i(pm-qn)}{N}} J_p J_{-q} = \sum_{p\neq\{p_s\}} e^{\frac{2\pi ipm}{N}} J_p \sum_{q\neq\{p_s\}} e^{\frac{2\pi iqn}{N}} J_{-q} \simeq$$

$$\simeq \left(\varepsilon_m - g_{m,0}\sqrt{\frac{r}{N}}\right)\left(\varepsilon_n - g_{n,0}\sqrt{\frac{r}{N}}\right) \sim \varepsilon_m\varepsilon_n \ . \quad (68)$$

The third term $H_{m,n}^{(3)}$ within the same approximation reads as

$$\left(\frac{2}{\pi}N^{2-\gamma/2}\sin\theta\right)H_{m,n}^{(3)} = \sum_{p,q\neq\{p_s\}} e^{\frac{2\pi i(pm-qn)}{N}} \sum_{l=1}^{r}(2l-1)\left(J_{p-2l+1}J_{2l-1-q} - J_{p+2l-1}J_{-2l+1-q}\right) =$$

$$= \sum_{l=1}^{r}(2l-1)\left[e^{\frac{2\pi i(2l-1)(m-n)}{N}}\left(\varepsilon_m - g_{m,2l-1}\sqrt{\frac{r}{N}}\right)\left(\varepsilon_n - g_{n,2l-1}^*\sqrt{\frac{r}{N}}\right) - \right.$$

$$\left. - e^{-\frac{2\pi i(2l-1)(m-n)}{N}}\left(\varepsilon_m - g_{m,2l-1}^*\sqrt{\frac{r}{N}}\right)\left(\varepsilon_n - g_{n,2l-1}\sqrt{\frac{r}{N}}\right)\right] \simeq$$

$$\simeq 2i\varepsilon_m\varepsilon_n\sum_{l=1}^{r}(2l-1)\sin\left[\frac{2\pi}{N}(2l-1)(m-n)\right] \sim$$

$$\sim 2i\varepsilon_m\varepsilon_n \begin{cases} \frac{r(4r^2-1)}{3}\frac{2\pi(m-n)}{N}, & |m-n| \ll \frac{N}{r} \\ c\frac{N^2\text{sign}(m-n)}{(m-n)^2} + 2r - \frac{N}{\pi(m-n)}, & |m-n| \gg \frac{N}{r} \end{cases} \quad (69)$$

In the last case we assumed that sine of the large argument is more or less equivalent to $(-1)^l$. In both latter derived equations we again used (66) and neglected these terms with respect to $\varepsilon_m$.

The last term is given by the truncated initial hopping term which we do not split into the above four sums (60):

$$H_{m,n}^{(4)} = \sum_{p\neq\{p_s\}} \frac{e^{\frac{2\pi ip(m-n)}{N}}}{N} E_p = \frac{2}{\pi}N^{-\gamma/2}\sin\theta \sum_{k=r}^{N/2} \frac{\sin\left[\frac{2\pi}{N}(2k-1)(m-n)\right]}{2k-1} \ . \quad (70)$$

Again using the same approximation for sine of the large argument as $(-1)^k$ we will obtain

$$H_{m,n}^{(4)} \sim \frac{2}{\pi}N^{-\gamma/2}\sin\theta \begin{cases} \left(1 - r\frac{m-n}{N}\right) + c\frac{m-n}{N}, & |m-n| \ll \frac{N}{r} \\ \frac{c}{r} - \frac{2c}{N}, & |m-n| \gg \frac{N}{r} \end{cases} \quad (71)$$

The first bracket in the first case corresponds to the small argument of the sine, $k \leq N/[4\pi(m-n)]$, while the rest terms correspond to the large sine argument.

To sum up, in the coordinate basis at $2r$th step we have the following estimate for the renormalized Hamiltonian given in Eq. (27)

$$H_{m,n}(2r) \sim \varepsilon_m\delta_{mn} + \frac{\varepsilon_m\varepsilon_n}{N^{2-\gamma/2}\cos\theta} +$$

$$+ \begin{cases} \frac{2}{\pi}N^{-\gamma/2}\sin\theta\left(1 - r\frac{m-n}{N}\right) + \frac{i8\pi^2\varepsilon_m\varepsilon_n(m-n)r^3}{3N^{3-\gamma/2}\sin\theta} - (\varepsilon_m + \varepsilon_n)\frac{r}{N}, & |m-n| \ll \frac{N}{r} \\ \frac{2}{\pi}N^{-\gamma/2}\sin\theta\frac{c}{r} + \frac{i2\pi\varepsilon_m\varepsilon_n}{N^{2-\gamma/2}\sin\theta}\left(c\frac{N^2\text{sign}(m-n)}{(m-n)^2} + r\right) - \frac{\varepsilon_m + \varepsilon_n}{2\pi|m-n|}, & |m-n| \gg \frac{N}{r} \end{cases}, \quad (72)$$

with $1 \leq r \leq N/4$ and a certain unimportant constant $c$.

## C Extrapolation of the multifractality measures to the thermodynamic limit $N \to \infty$

Here we remind the standard extrapolation procedure for the spectrum of fractal dimensions (see, e.g., [13, 15, 16, 27, 39]) and for the fractal dimensions $D_q$ [37].

For the first one we express the multifractal spectrum $f(\alpha, N)$ at finite system size $N$

$$f(\alpha, N) = f(\alpha) + \frac{A_\alpha^{(1)}}{\ln N} + \frac{A_\alpha^{(2)}}{(\ln N)^2} + \dots , \qquad (73)$$

with certain constants $A_\alpha^{(k)}$ depending on $\alpha$. The latter expression can be derived using the definition Eq. (38) and extracted directly from the histogram over $\alpha$ [13, 15, 39, 40]. Here and further we stick to the simplest linear in $1/\ln N$ behavior, which is typical for the models with fractal eigenstates [13, 15, 17].

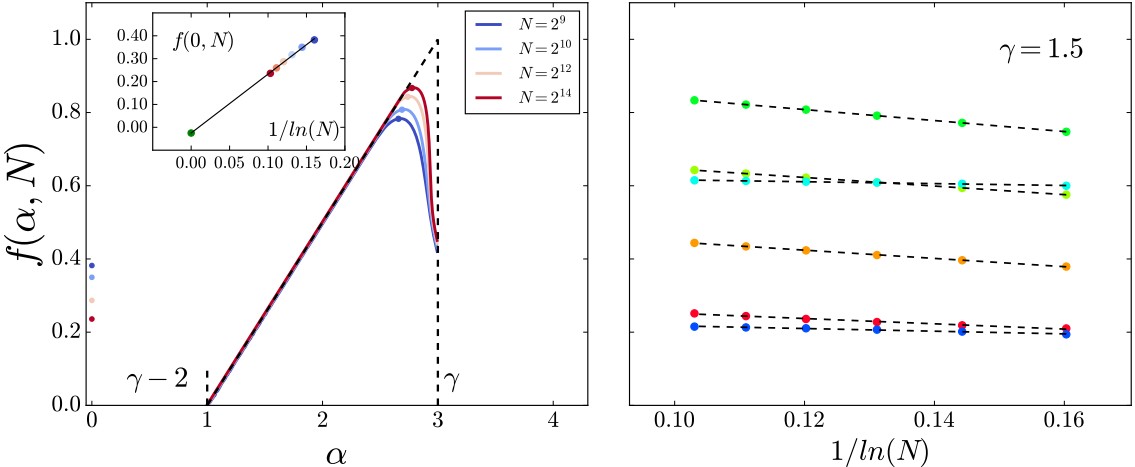

Figure 7: **Finite-size extrapolation of the multifractal spectrum** $f(\alpha)$ for 50 % mid-spectrum eigenstates and $\theta = 0.25$. (left) $f(\alpha, N)$ and its extrapolation for $\gamma = 3$, the inset shows the extrapolation of $f(0, N)$ vs $1/\ln N$, (right) extrapolation of $f(\alpha, N)$ vs $1/\ln N$ at $\gamma = 1.5$ for several values of $\alpha$. $f(\alpha)$ is extrapolated from $N = 2^9 - 2^{14}$ with 1000 disorder realizations for each $\gamma$ value.

The corresponding finite-size $f(\alpha, N)$ and extrapolated $f(\alpha)$ curves are given in Fig. 7 for 50 % of mid-spectrum states. As an additional marker of the extrapolation quality we check that the normalization condition, $\max_\alpha f(\alpha) = f(\alpha_0) = 1$, of the probability distribution $\mathcal{P}(\alpha)$ is satisfied.

The finite-size fractal dimension is defined by the formula (42) $D_q(N) = \ln I_q / (1 - q) \ln N$, with the generalized inverse participation ratio (IPR),

$$I_q = \sum_i |\psi_n(i)|^{2q} = c_q N^{(1-q)D_q} . \qquad (74)$$

In order to avoid the parasitic contributions from measure zero of special eigenstates we focus on the typical averaging of the IPR both over the disorder and the eigenstates

$$I_{q,typ} = e^{\langle \ln I_q \rangle} \sim N^{-(q-1)D_{q,typ}} \qquad (75)$$

and omit the subscript "typ" for brevity.

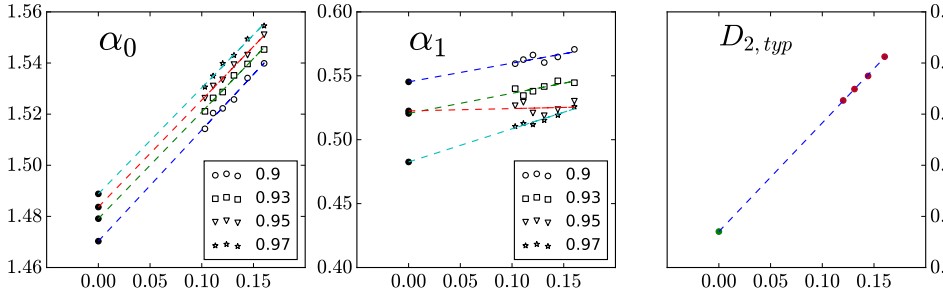

Figure 8: **Finite-size extrapolation of the fractal dimension** $\alpha_0$, $\alpha_1$ and $D_2$ for $\gamma = 1$, $\theta = 0.25$. $D_2$ is extrapolated from the same system sizes as $f(\alpha)$ in Fig. 7. Different symbols in the extrapolation of $\alpha_q$ correspond to different percentage $P$ of the deviation from the maximum of the function $f(\alpha_q) - q\alpha_q$ used for the extrapolation.

As the main contribution to IPR is given by the scaling exponent $D_q$ and the prefactor $c_q$ similarly to (73) one obtains

$$D_q(N) = D_q + \frac{(1-q)^{-1}\ln c_q}{\ln N} \; . \tag{76}$$

The extrapolation of $D_q(N)$ vs $1/\ln N$ extracted from $I_2$ and from $\alpha_0$ and $\alpha_1$ is shown in Fig. 8. Here in order to diminish finite $\alpha$-bin size for extracting $\alpha_q$ we fit $f(\alpha) - q\alpha$ close to its maximum with a parabolic fit and associate $\alpha_q$ with the maximal position of this fit. The fitting interval, $\alpha_- \leq \alpha \leq \alpha_+$, is determined by the deviation from the maximal value $f(\alpha_q) - q\alpha_q$ to

$$f(\alpha_\pm) - q\alpha_\pm = (f(\alpha_q) - q\alpha_q)\,P \; , \tag{77}$$

with the percentage $P = 0.9, 0.93, 0.95, 0.97$ shown in the legends of Fig. 8. One can straightforwardly see that 7 % change in $P$ affects the extrapolation by at most the same amount. The same procedure is done for $\alpha_0(N)$ and $\alpha_1(N)$ mentioned in Eqs. (39) and (44).

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
