# Peer review of "Localization and fractality in disordered Russian Doll model"

_SciPost Physics_

## Round 1 · Referee Report · Anonymous · 2022-4-15

Strengths

* A novel RG scheme is introduced in order to tame the strong energy divergences associated to the clean "band structure" in this model.
The RG yields an effective Hamiltonian that admits a determination of the fractal dimension of eigenstates in a certain range of scaling for the strength of the hopping terms.
* This adds another example of a random matrix ensemble that exhibits localized, non-ergodic, and ergodic states, of potential interest in the context of toy ("hopping on Fock space") versions of MBL.
* Analytical conclusions are supported with numerics

Weaknesses

(1) The biggest weakness of this paper is rather unfocused nature of the introduction. The opening paragraphs discuss the many-body Richardson and Russian Doll models, but while these "inspire" the study here, they have no direct relevance. The dirty RDM studied here is a problem in random matrix theory, and the basic conclusions (tunable regimes for ergodic, non-ergodic, and localized states) are more intelligible in that context. This is especially true because the many-body Richardson model and RDM exhibit a host of peculiar features (e.g. integrable quench dynamics, an infinite tower of BCS "ground" states) which have no relevance here.

Report

In this manuscript, the authors consider the eigenstate statistics of an infinite-ranged "hopping" model with onsite disorder, inspired in part by the Richardson (reduced BCS) and "Russian Doll" BCS models. As a many-body or reduced (Cooper) problem, the latter was shown to exhibit a novel "periodic RG", associated to the formation of a tower of BCS-type eigenstate solutions.

The emphasis here however is on the associated one-body (random-matrix) version of the dirty Russian Doll model (RDM), which does not exhibit the RG recurrence. This is an elaboration of the corresponding one-body Richardson (or "simplex") model studied previously. That model exhibits "localization" of all eigenstates but one. More precisely, in the one-body Richardson model all eigenstates take the form \psi_i = 1/(E - e_i), where E is the eigenenergy and e_i is the potential at site i. Most of the eigenstates are "localized" in the sense that IPR gives a multifractal exponent \tau(q) = 0, due to the infrared divergence for e_i < E < e_{i+1}.

By contrast, the addition of the imaginary, antisymmetric hopping term "h" in the present manuscript appears to allow for ergodic, fractal, and localized regimes, depending on how the hopping terms are scaled with N. The latter is controlled by an exponent \gamma. The authors use a version of the RDM RG [LeClair et al] to decimate out the highest-energy states in momentum space. Given the strongly divergent dispersion ~ cot(k), this is a sensible procedure, and produces an effective Hamiltonian that the authors then use to investigate fractality of the model. For 0 < \gamma < 2, they predict D = 1 - \gamma/2, a result that they also check numerically.

The conclusions of the paper appear sound, and the adapted RG method employed here is novel enough to warrant publication in SciPost.

Requested changes

(1) I suggest placing the work here in greater context earlier in the introduction, i.e. that the physics studied here is more analogous to the Rosenzweig-Porter random matrix model than the many-body Richardson or RD models. In particular it would be helpful to state that the limit cycle RG is NOT an ingredient here (despite the title).

(2) The analytical and numerical results for the fractal dimension D appear to be independent of the angle \theta (except for extreme cases, i.e. "Richardson" with \theta = 0). It would be helpful if the authors can explain concisely why this is.

(3) A perhaps general comment with respect to "localization" in this model and the one-body Richardson one is the following. In the latter, all eigenstates actually take the same form, \psi_i = 1/(E - e_i), where E is the eigenenergy and e_i is the potential at site i. There is no dependence on the hopping in this case, except indirectly through the energy eigenvalue E. If we "lexicographically order" the energies e_1 < e_2 < .. < e_N, then these eigenstates are power-law peaked with a center that merely drifts linearly with E. Moreover, these "localized" states show power-law Chalker scaling, since the states with nearby energies sit on nearby e_i's. This is very different from a true Anderson insulator, where the Chalker correlator is ~ 0 for energy differences less than the typical localization volume level-spacing. I would therefore prefer to refer to such states as "frozen" rather than localized, where freezing refers to \tau(q) = 0 for q \geq 1, but the spatial structure can consist of rare peaks that are separated by arbitrarily large distances.
Such states also show Chalker scaling.
(see e.g. Chou and Foster, PHYSICAL REVIEW B 89, 165136 (2014)).

Are the localized states obtained here (for \gamma > 2) also "frozen" in this sense?

  • validity: high
  • significance: good
  • originality: high
  • clarity: good
  • formatting: good
  • grammar: reasonable

Author:  Ivan Khaymovich  on 2022-06-02  [id 2549]

(in reply to Report 1 on 2022-04-15)

Report 1 of the Referee Report In this manuscript, the authors consider the eigenstate statistics of an infinite-ranged "hopping" model with onsite disorder, inspired in part by the Richardson (reduced BCS) and "Russian Doll" BCS models. As a many-body or reduced (Cooper) problem, the latter was shown to exhibit a novel "periodic RG", associated to the formation of a tower of BCS-type eigenstate solutions. The emphasis here however is on the associated one-body (random-matrix) version of the dirty Russian Doll model (RDM), which does not exhibit the RG recurrence. This is an elaboration of the corresponding one-body Richardson (or "simplex") model studied previously. That model exhibits "localization" of all eigenstates but one. More precisely, in the one-body Richardson model all eigenstates take the form $\psi_i = 1/(E - e_i)$, where $E$ is the eigenenergy and $e_i$ is the potential at site i. Most of the eigenstates are "localized" in the sense that IPR gives a multifractal exponent $\tau(q) = 0$, due to the infrared divergence for $e_i < E < e_{i+1}$. By contrast, the addition of the imaginary, antisymmetric hopping term "h" in the present manuscript appears to allow for ergodic, fractal, and localized regimes, depending on how the hopping terms are scaled with N. The latter is controlled by an exponent $\gamma$. The authors use a version of the RDM RG [LeClair et al] to decimate out the highest-energy states in momentum space. Given the strongly divergent dispersion $\sim cot(k)$, this is a sensible procedure, and produces an effective Hamiltonian that the authors then use to investigate fractality of the model. For $0 < \gamma < 2$, they predict $D = 1 - \gamma/2$, a result that they also check numerically.

We thank the referee for careful reading of our manuscript and for summarizing our results. We would like just to mention a couple of points which seem to be a bit unclear:

(a) the RG recurrence mentioned by the referee does not break down in the single-particle (random matrix) sector of the disorder-free Russian Doll model. It is the diagonal disorder, which breaks down the structure of renormalized Hamiltonian (please see the reply (c) to the question (3) and [34] for more details). (b) The RG used in our manuscript has the origin from [7-8], but is not just a version of it: the main idea of decimating out the highest-energy states is the same in both, but in our case, we have - decimated out the states which are non-ergodic in the momentum space, - not observed any RG recurrence, and therefore - had to introduce the stopping criterion for the RG procedure based on the optimization of the broadening-based fractal dimension. In the revised version of the manuscript we have emphasized both above issues more clearly.

Weaknesses (1) The biggest weakness of this paper is rather unfocused nature of the introduction. The opening paragraphs discuss the many-body Richardson and Russian Doll models, but while these "inspire" the study here, they have no direct relevance. The dirty RDM studied here is a problem in random matrix theory, and the basic conclusions (tunable regimes for ergodic, non-ergodic, and localized states) are more intelligible in that context. This is especially true because the many-body Richardson model and RDM exhibit a host of peculiar features (e.g. integrable quench dynamics, an infinite tower of BCS "ground" states) which have no relevance here. Requested changes (1) I suggest placing the work here in greater context earlier in the introduction, i.e. that the physics studied here is more analogous to the Rosenzweig-Porter random matrix model than the many-body Richardson or RD models. In particular it would be helpful to state that the limit cycle RG is NOT an ingredient here (despite the title).

We thank the Referee for the suggestion, but we kindly disagree with his/her vision on the relevance of many-body RDM and Richardson models to the considered problem. Indeed, the presence of an infinite tower of BCS "ground" states in RDM has immediate similarity to the hierarchy of divergent single-particle levels which are both ergodically delocalized in the coordinate basis and their number scales in the same way with $N$. Bethe ansatz integrability of the many-body Richardson and RDM models is also relevant to the single-particle sector as already in the Richardson model this integrability does lead to localization, but cannot avoid level repulsion similar to the Gaussian random matrix case. Therefore already on these simple examples one can see the relevance of the interplay of Bethe-ansatz integrability, disorder, and time-reversal symmetry breaking which (unlike Richardson model) leads to the non-ergodic delocalization at $\gamma<2$. The question of the many-body sectors of RDM is still an open issue and should be addressed in the further investigations. Having all this in mind, we prefer to keep the structure of the introduction, only slightly modifying the accents there.

(2) The analytical and numerical results for the fractal dimension D appear to be independent of the angle $\theta$ (except for extreme cases, i.e. "Richardson" with $\theta = 0$). It would be helpful if the authors can explain concisely why this is.

We thank the Referee for the question. Indeed, our results do not depend on $\theta$ for any non-zero values of this parameter. Formally from Eqs. (32-33) one can see that for finite-size models the number of RG steps $r_{opt}$ and the corresponding broadening $\Gamma_n(r_{opt})$ have the common crossover $\sin\theta \sim W N^{-1+\gamma/2}\ll 1$ beyond which $r_{opt}<1$ and $\Gamma_n<\delta$. The later returns us back to the Richardson's model as one does not need to make a single RG step and obtains localization. The origin of this finite-size crossover is given by the divergence of the disorder-free spectrum (8) at least at the largest energy scale $E_{2k+1=1}$. As soon as $E_1 \leq W$ the disorder hybridize it with the rest disorder-free levels and the derivation of the RG in the momentum space does not work anymore (see the discussion after Eq. (8) in the end of the section 3). In the revised version of the manuscript, we have slightly reformulate this discussion in order to emphasize the above points.

(3) A perhaps general comment with respect to "localization" in this model and the one-body Richardson one is the following. In the latter, all eigenstates actually take the same form, $\psi_i = 1/(E - e_i)$, where $E$ is the eigenenergy and $e_i$ is the potential at site i. There is no dependence on the hopping in this case, except indirectly through the energy eigenvalue E. If we "lexicographically order" the energies $e_1 < e_2 < .. < e_N$, then these eigenstates are power-law peaked with a center that merely drifts linearly with E. Moreover, these "localized" states show power-law Chalker scaling, since the states with nearby energies sit on nearby $e_i$'s. This is very different from a true Anderson insulator, where the Chalker correlator is $\sim 0$ for energy differences less than the typical localization volume level-spacing. I would therefore prefer to refer to such states as "frozen" rather than localized, where freezing refers to $\tau(q) = 0$ for $q \geq 1$, but the spatial structure can consist of rare peaks that are separated by arbitrarily large distances. Such states also show Chalker scaling. (see e.g. Chou and Foster, PHYSICAL REVIEW B 89, 165136 (2014)). Are the localized states obtained here (for $\gamma > 2$) also "frozen" in this sense?

We thank the Referee for pointing out this issue of Chalker scaling and possible frozen nature of the localized phase. In order to reply to the Referee's question, we need to mention at least 3 points: (a) First of all, if one speaks about the localized phase, $\gamma>2$, the Russian Doll, the Richardson, and the Rosenzweig-Porter models show identical results given by the perturbation theory (see [12] for the discussion of correlations and [16] for the Chalker-like scaling). In this case, all three models show power-law localization in "lexicographically ordered" energies $e_1 < e_2 < .. < e_N$ with the power $a=2$, see, e.g., Fig. 6 of the revised version. Thus, the corresponding Chalker scaling gives in this case for the overlap correlation function $K(\omega)\sim (\delta/\omega)^2$. (b) Second, generally in all long-range models with power-law Anderson localization (see, e.g., the well-known power-law random banded matrices (PLRBM) or less known Burin-Maksimov model in [12, 26, 27]) the critical exponents $\tau_q$ are non-zero for $q<1/(2a)$, where $a>1$ is the power of the localization, $|\psi_n(r)|\sim 1/|n-r|^a$. The same is true for the above-mentioned Rosenzweig-Porter model at $\gamma>2$, where $\tau_q>0$ for $q<1/\gamma$. In all these cases formally the localized states can be called frozen, but non-zero critical exponents appear only somewhere at $q<1/2$. The same is true for the disordered Russian Doll model as in the localized phase it shows the same structure as the Rosenzweig-Porter model.

(c) Another important issue is the difference between Richardson or Rosenzweig-Porter models and, e.g., PLRBM and Russian Doll model in terms of the Chalker's ansatz in the non-ergodic extended phase. Both in the Richardson and Rosenzweig-Porter models the wave-function structure is given by the diagonal disorder in non-ergodic phases, due to the absence of any other spatial structure in the model. In the Richardson's model it appears due to the eigenstate structure $\psi_i = 1/(E - e_i)$, given by the Bethe ansatz solution and mentioned by the Referee. For the Rosenzweig-Porter model the similar structure of the Breit-Wigner form appears due to the self-averaging in the broadening $\Gamma_n$ and its independence of the space index $n$. Unlike this, in the PLRBM model there is the competing structure of the power-law decaying hopping terms (in the usual order of sites) and the one of the diagonal elements (in the lexicographic order of sites mentioned by the Referee). It is this competition of two reordered spaces, which make the critical states multifractal in the PLRBM and introduces a non-trivial power to the Chalker's scaling. In the case of the Russian Doll model, this competition is present, but it is given minimally by the presence of the sign-distance dependence ${\rm sign}(d(m,n))$ in the TRS-breaking term in (2). But in any case the presence of such a competition immediately breaks down the RG considered in [7-8] and leads to the power-law decaying hopping in the effective Hamiltonian, the second case in (34). This changes the wave-function structure of the Richardson's model, $\psi_i = 1/(E - e_i)$ at $\gamma<2$ with respect to the Russian Doll model and therefore should affect the corresponding Chalker scaling. We leave the investigation of the later for the further study.

In the revised manuscript we have added the discussion of the above issues in a brief form.

Attachment:

Russian_Doll_v7_SciPost.pdf

---

## Round 2 · Referee Report · Anonymous (Referee 2) · 2022-8-1

Strengths

The main strength is convincing renormalization group treatment of the model by two different methods, which well agrees with outcome of the exact diagonalization

Weaknesses

Weakneses are mainly stylistic: the English is a bit lame to the extent that the precise meaning of some places becomes vague.

Report

I like the content of this work - a carefully executed renormalization in an all-to-all "Russian Doll'' model with diagonal disorder and broken time-reversal symmetry. Evidence for accuracy of the RG is overwhelming: two complementary methods give the same effective Hamiltonian which eventually can be treated (at least, for establishing the multifractality properties of eigenfunctions) largely to the same extent as the paradigmatic Porter-Rosenzweig model.

Unfortunately exposition is frequently vague, and I believe poor usage of English grammar contributes to vaguesness. Though I provide below a few suggestions for improvement, I would further recommend the authors to ask someone with a good command of English grammar to edit the paper throughout.

Requested changes

The research quality is high, but the exposition needs a certain editing before acceptance. I give below a few suggestions for improvement

(1) top of page 7:
hopping between plane waves
-->
hopping between states in the plane wave basis

(2) after eq.(24): the above approximation works leading to ...
and the difference between ... and ... at most of the order
-->
the above approximation works WELL leading to ... the difference between ... and ... and IS at most of the order

(3) top of page 11:
and inverse this matrix
-->
and inverTS this matrix.

(4) with one-sided divergence
-->
with one sided unbounded growth

(5) where Ep>1 are large and positive Ep<p∗ > 0
-->
where Ep>1 are large and positive.

(5) page 13:
"with a bandwidth b" - note that b is not defined in eq. 34 (I suppose it W/\Gamma?)

footnote 4 " there are some investigations which might have multifractal wave functions'' - it sounds nonsensical ...
please try to reformulate.

(6) eq.(40): note that such a symmetry was originally discovered in:
AD Mirlin, YV Fyodorov, A Mildenberger, F Evers
Exact relations between multifractal exponents at the Anderson transition
Physical review letters 97 (4), 046803 (2006)

Please cite the original reference, not only the review [37].

(7) In conclusions: " we ... confirm the subleading character of the
approximations'' --> we confirm that approximations we used are valid to the leading order, and all subsequent corrections are subleading (and write explicitly in which parameter)

  • validity: high
  • significance: high
  • originality: good
  • clarity: ok
  • formatting: -
  • grammar: below threshold

Author:  Ivan Khaymovich  on 2022-08-29  [id 2770]

(in reply to Report 1 on 2022-08-01)

** Report 1 of the Referee **
** Report **
** I like the content of this work - a carefully executed renormalization in an all-to-all "Russian Doll'' model with diagonal disorder and broken time-reversal symmetry. Evidence for accuracy of the RG is overwhelming: two complementary methods give the same effective Hamiltonian which eventually can be treated (at least, for establishing the multifractality properties of eigenfunctions) largely to the same extent as the paradigmatic Porter-Rosenzweig model. **
** Unfortunately exposition is frequently vague, and I believe poor usage of English grammar contributes to vagueness. Though I provide below a few suggestions for improvement, I would further recommend the authors to ask someone with a good command of English grammar to edit the paper throughout. **

We thank the referee for the careful reading of the manuscript. \textit{As per the referee's recommendation, we have carried out a thorough editing of the paper to make the exposition clearer. Below, we list down corrections made as per the referee's specific comments:}

** Requested changes **
** The research quality is high, but the exposition needs a certain editing before acceptance. I give below a few suggestions for improvement **
** (1) top of page 7:
hopping between plane waves
-->
hopping between states in the plane wave basis **

We have replaced the phrase by "Indeed, the disorder term ... in the momentum-space basis... plays a role of the scattering between plane waves (or hopping),..."

** (2) after eq.(24): the above approximation works leading to ...
and the difference between ... and ... at most of the order
-->
the above approximation works WELL leading to ... the difference between ... and ... and IS at most of the order **

We have followed the recommendation of the referee.

** (3) top of page 11:
and inverse this matrix
-->
and inverTS this matrix. **

We have followed the recommendation of the referee.

** (4) with one-sided divergence
-->
with one sided unbounded growth **

In the correspondent place and in other ones we have replaced spectral divergence by the spectral unbounded growth.

** (5) where $E_{p<p^*} \gg 1$ are large and positive $E_{p<p^*} > 0$
-->
where $E_{p<p^*} \gg 1$ are large and positive. **

We have followed the recommendation of the referee.

** (5) page 13:
"with a bandwidth b" - note that b is not defined in eq. 34 (I suppose it $W/\Gamma$?) **

We have followed the recommendation of the referee and added the definition of the effective parameter $b$.

** footnote 4 " there are some investigations which might have multifractal wave functions'' - it sounds nonsensical ...
please try to reformulate. **

We have rephrased the footnote 4 according to the referee's comment.

** (6) eq.(40): note that such a symmetry was originally discovered in:
AD Mirlin, YV Fyodorov, A Mildenberger, F Evers
Exact relations between multifractal exponents at the Anderson transition
Physical review letters 97 (4), 046803 (2006)
Please cite the original reference, not only the review [37]. **

We have added the reference to the above paper and mentioned that the corresponding symmetry was originally discovered there.

** (7) In conclusions: " we ... confirm the subleading character of the
approximations'' --> we confirm that approximations we used are valid to the leading order, and all subsequent corrections are subleading (and write explicitly in which parameter) **

We have reformulated the above phrase in the conclusions accordingly.

---

## Round 2 · Author Response

Dear Prof. Sarang Gopalakrishnan,

Thank you for communicating to us the Referee report on our manuscript entitled "Localization and fractality in disordered Russian Doll model". We would like to resubmit the article for further consideration for SciPost Physics.
We would like to thank the Referee for taking the time to have a careful read of our manuscript and for the report. The Referee communicated a really positive assessment of our manuscript, and she/he mentions that our conclusions "appear sound, and the adapted RG method employed here is novel enough to warrant publication in SciPost."

We provide a detailed reply to the Referee critique and attach it to the Referee report. We include the full reports in bold text, and comment in normal font to all the points indicating also the related changes made in the manuscript. We have also attached the revised manuscript with the performed changes highlighted in red to the reply to the referee report, so that the they are easier to spot.

Sincerely yours,
Vedant Motamarri, Alexander S. Gorsky, and Ivan M. Khaymovich

---

## Round 2 · List of Changes

• The references [25, 34, 36, 40-47] have been added. The other references have been shifted accordingly.
  • We have slightly changed the accents in the introduction and in the abstract.
  • We have added a brief discussion of the power-law-like localization and its relation to the frozen multifractal states and Chalker scaling in the end of Sec. 2.
  • We have slightly reformulate this discussion after Eq. (8) in order to emphasize the finite-size crossover at $\sin \theta \sim W N^{-1+\gamma/2}$.
  • We have added a discussion of many-body sectors of the models after the previous item discussion.
  • In addition, we have added the numerical results on the spectrum of fractal dimensions $f(\alpha)$, on the level $r$-statistics across the spectrum, and on the wave-function decay in Figs. 4-6 as well as the corresponding analytical calculations (35-37).

---

## Editorial Decision

resubmitted